# Antioxidant Enzymes in Cancer Cells: Their Role in Photodynamic Therapy Resistance and Potential as Targets for Improved Treatment Outcomes

**DOI:** 10.3390/ijms25063164

**Published:** 2024-03-09

**Authors:** Wachirawit Udomsak, Malgorzata Kucinska, Julia Pospieszna, Hanna Dams-Kozlowska, Waranya Chatuphonprasert, Marek Murias

**Affiliations:** 1Department of Toxicology, Poznan University of Medical Sciences, ul Rokietnicka 3, 60-608 Poznan, Poland; wachirawit5414@gmail.com (W.U.); kucinska@ump.edu.pl (M.K.); julia.pospieszna@gmail.com (J.P.); 2Department of Cancer Immunology, Poznan University of Medical Sciences, ul. Garbary 15, 61-866 Poznan, Poland; hdamskozlowska@ump.edu.pl; 3Department of Diagnostics and Cancer Immunology, Greater Poland Cancer Centre, ul.Garbary 15, 61-866 Poznan, Poland; 4Faculty of Medicine, Mahasarakham University, Maha Sarakham 44000, Thailand; waranya.c@msu.ac.th; 5Center for Advanced Technology, Adam Mickiewicz University, ul. Uniwersytetu Poznańskiego 10, 61-614 Poznan, Poland

**Keywords:** photodynamic therapy, superoxide dismutase, catalase, glutathione redox cycle, heme oxygenase-1, antioxidant inhibitors, cancer

## Abstract

Photodynamic therapy (PDT) is a selective tumor treatment that consists of a photosensitive compound—a photosensitizer (PS), oxygen, and visible light. Although each component has no cytotoxic properties, their simultaneous use initiates photodynamic reactions (PDRs) and sequentially generates reactive oxygen species (ROS) and/or free radicals as cytotoxic mediators, leading to PDT-induced cell death. Nevertheless, tumor cells develop various cytoprotective mechanisms against PDT, particularly the adaptive mechanism of antioxidant status. This review integrates an in-depth analysis of the cytoprotective mechanism of detoxifying ROS enzymes that interfere with PDT-induced cell death, including superoxide dismutase (SOD), catalase, glutathione redox cycle, and heme oxygenase-1 (HO-1). Furthermore, this review includes the use of antioxidant enzymes inhibitors as a strategy in order to diminish the antioxidant activities of tumor cells and to improve the effectiveness of PDT. Conclusively, PDT is an effective tumor treatment of which its effectiveness can be improved when combined with a specific antioxidant inhibitor.

## 1. Introduction

Photodynamic therapy (PDT) is a selective tumor treatment that has been approved by the United States Food and Drug Administration (FDA) since the late 1990s [1,2,3]. PDT generates cytotoxic reactive oxygen species (ROS), including singlet oxygen (^1^O_2_), hydroxyl radicals (HO^•^), and superoxide anions (O_2_^•–^), in the presence of oxygen, through various types of photodynamic reactions (PDRs). In order to initiate a PDR, a photosensitizer (PS) needs to be activated by visible light of a specific wavelength. The light excites the electron of the PS in the ground state into an excited singlet state. Then, the excited PS can return to its ground state, producing photon emission (fluorescence), or convert to a triplet state through intersystem crossing. Eventually, the excited triplet state PS transfers its electrons to the formation of free radical species (e.g., hydroxyl radicals and superoxide anions) and oxidizes the subcellular substrates as type I PDR electron transfer. In the meantime, the excited triplet state PS can transfer energy to oxygen molecules for the formation of singlet oxygen and can cause cell death via type II PDR energy transfer. It is important to note that both type I and II PDRs are oxygen-dependent reactions that can proceed simultaneously depending on PS properties and the cellular oxygen level [4,5,6]. Recently, PDRs have been additionally classified into types III and IV based on the direct activation of the PS. After activation, types III and IV PDRs can immediately exert cytotoxicity and require neither additional reactions nor oxygen molecules [7,8,9,10].

The anticancer effects of PDT-induced therapy are classified into three mechanisms depending on the properties of the PS, light dosage, and tumor environment, as follows: (1) irreversible direct cell killing—the cytotoxic ROS directly injures organelles of cancer cells, which results in the induction of several cell death pathways, including conventional (apoptosis, autophagy, and necrosis) and “non-conventional” pathways, such as ferroptosis, mitotic catastrophe, paraptosis, pyroptosis, parthanatos, necroptosis, and immunogenic cell death (ICD); (2) vasculature damage—the PS, localized in the endothelial cells, destroys blood vessel walls, resulting in blood supply interruption; and (3) inflammation and immune responses—PDT can trigger inflammatory and immune responses by recruiting inflammatory mediators, e.g., cytokines, leukotrienes, and tumor necrosis factors [5,11,12,13,14,15,16]. PDT surpasses conventional chemotherapy in various aspects, especially in specificity and efficacy. The enhanced permeability and retention (EPR) effect, resulting from the leaky tumor vasculature, combined with the overexpression of low-density lipoprotein receptors, enables the passive delivery of the PS to tumor sites [17]. As a result of PS exposure to visible light of a specific wavelength, precise radiation is delivered to the tumor niche. This, in turn, triggers a PDR, producing cancer cell-specific ROS. Moreover, PDT often shows minimal cross-resistance with chemoresistance, radioresistance, or other PSs [18]. Combining PDT with other therapeutic modalities, such as surgery, chemotherapy, radiotherapy, and immunotherapy, can synergistically improve treatment outcomes [19,20,21,22].

Although the first PS was clinically introduced in 1978, many aspects of PDT still need refinement, especially regarding the hypoxic conditions in the tumor environment and the specific wavelength required for the excitation of the PS [2,23]. Despite PDT having many advantages, some factors limit its widespread use in clinical practice. First and foremost, in the classical understanding of PDT, oxygen is necessary for the cytotoxic effect to occur during PDT. Tumor hypoxia is not only considered the cause of cell resistance against chemo- or radiotherapy but also lowers the effectiveness of PDT, whether in vitro, pre-clinical, or clinical. Hypoxia is a well-known tumor feature and many regions of the tumor show deficient oxygen levels (less than 5 mmHg partial pressure of oxygen (pO_2_); 5 mmHg corresponds to approximately 0.7% O_2_ in the gas phase or 7 μM in solution) [24]. The effectiveness of PDT, especially that which is mainly mediated by oxygen-dependent type II PDRs, is eventually diminished due to a lower ROS production in the hypoxic environment of tumor cells [20]. Moreover, the wavelength of visible light must be specific to the PS to initiate the PDR. Nonetheless, the optical window to penetrate the tissue in PDT has been limited to only between 600 and 800 nm (red to deep red). More than 800 nm of light does not provide enough energy to activate the PS and produce ROS. At the same time, up to 600 nm light could be absorbed by water, leading to the minimization of the penetration of irradiation through the tissue, causing ineffectiveness against deep tumor sites. The light of a wavelength between 700 and 800 nm can penetrate deep into the tissue, approximately 1 cm, whereas 600 nm is limited to only 0.5 cm [25]. The absorption spectra of significant chromophores in tissues, i.e., cytochromes, melanin, water, deoxyhemoglobin, and oxyhemoglobin, were reported by Plaetzer’s group [26]. It is important to note that hemoglobin and oxyhemoglobin exhibit similar absorption wavelengths to some photosynthesizers used in PDT. Certain studies suggest that PDT may elevate the production of methemoglobin and deoxygenated hemoglobin, particularly under the conditions of hemolysis [27,28,29,30]. Consequently, there is no ideal visible light for all PDT indications. To select the optimal wavelength, one should consider the characteristics of the PS, such as its fluorescence excitation and action spectra, dosimetry factors like light dosage, exposure time, delivery mode, fluence rate, and disease attributes, including accessibility, location, size, and tissue specificity [19,31,32,33,34].

Another issue affecting PDT’s efficacy is the compensatory pathways activated in cancer cells in response to treatment. Cancer cells deploy different cytoprotective mechanisms, including the control of (1) antioxidant molecules levels, (2) enzymes in ROS detoxification, and (3) expression of specific genes encoding proteins in response to PDT [35]. Typically, cells balance between cellular oxidants and antioxidants under physiological conditions through redox homeostasis, which the ROS control at sub-micromolar levels. Paradoxically, cancer cells produce elevated ROS levels as an adaptive mechanism, sustaining a consistently pro-oxidative state that promotes tumor initiation and progression [36]. However, if ROS levels are extremely high, they can lead to tumor cell death. Therefore, this adaptive mechanism develops antioxidant systems against excessive ROS and becomes redox resetting, contributing resistance to various ROS-mediated tumor therapies, including PDT (Figure 1) [37,38].

To date, targeting redox proteins has been proposed as a potential strategy to enhance the therapeutic efficacy of various anticancer modalities, such as chemotherapy, radiotherapy, and PDT. However, it should be emphasized that antioxidant therapeutic strategies in cancer are quite complex, as they may involve both the supplementation of and the reduction in antioxidant activity in cancer cells. The beneficial effects of antioxidants are linked to their ability to mitigate the side effects of excessive free radical formation during anticancer treatments. Additionally, free radicals/ROS play crucial roles in tumorigenesis and cancer development. However, in the context of ROS-mediated therapy, decreasing antioxidant activity is particularly interesting, and certain selected agents or naturally occurring compounds have also been tested. In this review, we provide an in-depth analysis of the cytoprotective mechanisms of detoxifying ROS enzymes, generally considered as antioxidant enzymes that interfere with PDT-induced cell death. This paper will focus on three main antioxidant enzymes, as follows: superoxide dismutase, catalase, and the glutathione redox cycle. The relationships between these enzymes are depicted in Figure 2. Moreover, in this review, we also focused on heme oxygenase, since several reports showed the promising results for combining its inhibitor with PDT.

## 2. Antioxidant Enzymes Responsible for ROS-Mediated Treatment Resistance

### 2.1. Superoxide Dismutase (SOD)

Superoxide dismutases are the central antioxidant enzymes in the oxidoreductase group of the ROS defense system responsible for the dismutation of a rapidly reactive ROS, the superoxide anion (Figure 2) [42]. The superoxide anion is generated by various intracellular sources, including the mitochondrial electron transport chain (ETC), NADPH oxidase (NOX), nitric oxide synthase (NOS), lipoxygenase, xanthine oxidase (XO), cyclooxygenase (COX), cytosolic xanthine oxidase, cytochrome P450 mono-oxygenases, and mitochondrial enzymes (Mits) [40,43]. Under the pathological conditions of cardiovascular diseases, the superoxide anion causes oxidative stress and redox imbalance toward ROS/reactive nitrogen species (RNS)-related pathways. The simplest way is the protonation of the superoxide anion to a hydroperoxyl radical (HO_2_^•^), a strong oxidant that can further trigger lipid peroxidation [44,45,46]. An antioxidant, nitric oxide (NO), rapidly reacts with the superoxide anion, producing the strong oxidant, peroxynitrite (ONOO^−^), disrupting both endothelial and mitochondrial functions, leading to atherosclerosis, diabetes, hypercholesterolemia, hypertension, and aging [47,48]. The superoxide anion facilitates the generation of the highly toxic hydroxyl radical via reducing ferric iron (Fe^3+^) to ferrous iron (Fe^2+^) via the Haber–Weiss reaction, before reacting with H_2_O_2_ and turning back to Fe^3+^ via the Fenton reaction [46,49]. SODs are responsible for removing ROS via the catalysis of the superoxide anion to H_2_O_2_ involved with catalases, thioredoxin-peroxiredoxin, and glutathione. The scavenging mechanism of SODs has been called the “ping-pong” mechanism as it relates to both the reduction and the oxidation of catalytic metals, such as copper (Cu) or manganese (Mn) at the active site of the enzyme [42,50].

There are three distinct isoforms of SODs in mammals (Table 1). SOD1 (copper- and zinc-containing SOD) is localized in intracellular cytoplasmic compartments, including the cytoplasm, mitochondrial intermembrane space, lysosomes, peroxisomes, and nucleus. It regulates NOX2 activity, increasing endosome superoxide anions. Then, it catalyzes the dismutation of superoxide anions at the endosomal surface and produces localized hydrogen peroxide (H_2_O_2_), leading to the activation of the transcriptional factor NF-kB (nuclear factor kappa-light-chain-enhancer of activated B cells) [51]. However, SOD1 in the nucleus is essential for protecting against the oxidative DNA damage caused by H_2_O_2_. It acts as a transcriptional factor that regulates the expression of oxidative response genes, resulting in oxidative stress resistance [52]. SOD2 (manganese-containing SOD), specifically localized in the mitochondria matrix, generates H_2_O_2_ signaling inside the mitochondria, regulating blood vessel formation, cell differentiation, and pulmonary hypertension development [42,53]. SOD3 (copper- and zinc-containing SOD) is localized in the vascular extracellular space, extracellular matrix, cell surface, plasma, and extracellular fluids (i.e., lungs, blood vessels, kidneys, uterus, and heart). It plays a vital role in maintaining the redox homeostasis of the tissues from oxidative and inflammatory damage. It is also well known to possess anti-angiogenic, anti-inflammatory, antichemotactic, antiproliferative, and immunomodulatory properties [42,54,55,56].

In unhealthy cells, SODs impair the apoptosis induction of tumor cells by interrupting the hypochlorous acid (HOCl) and NO/peroxynitrite signaling pathways. Tumor cells require a high production rate of extracellular superoxide anions and H_2_O_2_ to maintain their development and progression [57,58,59]. On the other hand, H_2_O_2_ is a primary substrate for HOCl generation in the pathway of apoptosis-mediated PDT, namely HOCl signaling (Figure 3). HOCl is a potent oxidizing agent and, under physiological conditions, is mainly generated by the interaction between H_2_O_2_ and the chloride anion (Cl^−^), catalyzed by peroxidases (PODs), such as myeloperoxidase (MPO) or the peroxidase domain of dual oxidase (DUOX) [60,61]. After interaction with extracellular superoxide anions in tumor cells, HOCl yields an apoptosis-induced hydroxyl radical, which results in selective anti-tumor activity. Hence, the anti-tumor activity of HOCl signaling correlates with the hydroxyl radical level, which depends on the superoxide anion and HOCl levels in tumor cells. Hydroxyl radicals can reach the cell membrane and trigger lipid peroxidation, specifically in the membrane of tumor cells. Then, apoptosis is activated in the mitochondrial pathway relating to caspase-9 and -3 activity [62,63,64,65,66]. Furthermore, HOCl modulates immune response through an accumulation of neutrophils and the induction of tumor necrotic factor α (TNFα) production in peripheral blood mononuclear cells, contributing to inflammatory responses [67,68].

Another trail of apoptosis-mediated PDT is the NO/peroxynitrite signaling pathway (Figure 3). NO plays a vital role in both metabolic and cardiovascular balance, boasting properties that promote vasodilation, combat inflammation, and prevent thrombosis, as well as acting as an antioxidant [69]. Yet, when NO swiftly interacts with superoxide anions, it results in the formation of a potent oxidant known as peroxynitrite. This formidable oxidant has the capacity to modify a range of molecules. Among its targets are the heme of soluble guanylate cyclase, various lipids, and the endothelial NOS cofactor BH4. Such interactions can result in mitochondrial dysfunction, escalated ROS production, and lipid peroxidation [42]. Furthermore, peroxynitrite can be protonated to peroxynitrous acid (ONOOH) before decomposing to nitrogen dioxide (NO_2_) and the apoptosis-inducing hydroxyl radical. This protonation can specifically occur in the membrane of tumor cells, due to the acidic conditions in the presence of the proton pump of tumor cells. The in-depth mechanisms of HOCl and NO/peroxynitrite signaling have been thoroughly explained by Bauer and co-workers [70,71].

All isoforms of SODs are responsible for scavenging extracellular superoxide anions and converting them into H_2_O_2_. Next, H_2_O_2_ can be converted to H_2_O by catalase, peroxiredoxins (Prxs), or glutathione peroxidases (GPxs). Thus, they interfere with HOCl signaling by lowering the extracellular superoxide anion levels, leading to diminishing hydroxyl radical production and HOCl-mediated apoptosis induction (Figure 4) [71]. Likewise, NO/peroxynitrite signaling has been intercepted by SODs by preserving NO within the endothelium and preventing its interaction with superoxide anions, leading to a decrease in peroxynitrite formation. These mechanisms imply the evolution of free radical resistance within ROS/NOS apoptotic signaling pathways in tumor cells associated with SODs [42,72]. Bauer et al. reported that HOCl-mediated apoptosis induction was entirely inhibited by SOD2 treatment in gastric carcinoma MKN-45 cells [62]. They also confirmed that extracellular superoxide anion concentration in tumor cells could be increased by SOD inhibition. Likewise, pretreatment with a neutralizing antibody against SOD increased the cell sensitivity to HOCl-mediated apoptosis by increasing the HOCl concentration in a model of murine melanoma B16F10 cells. It suggested that SOD inhibition increased both the production of available superoxide anions and HOCl signaling [62]. Liu and co-workers discovered that the elevated protein level of SOD2 was associated with lymph node metastasis in a tongue squamous cell carcinoma (TSCC) model [73,74]. Among three TSCC cell lines, UM1 possessed the highest migration and invasion abilities correlated with the highest SOD2 protein level and activity. In addition, the H_2_O_2_ production and proliferation rate of UM1 cells were significantly higher than UM2 cells, another TSCC cell line with lower migration and invasion abilities. On the contrary, the aggressiveness of TSCC was reduced after knockdown of the expression of the SOD2 gene. UM1 transfected with SOD2 shRNA showed lower migration and invasion abilities, H_2_O_2_ production, and cell proliferation rate. Therefore, the migration and invasion abilities of TSCC were dependent on the production of H_2_O_2_ that was regulated by SOD2 [75].

Various evidence indicates the impact of SODs on the effectiveness of PDT. An immortalized T lymphocyte cell line, Jurkat, characterized by an enhanced SOD2 activity due to transfection with a gene encoding wild-type SOD2, showed a significant decrease in the generation of superoxide, activation of caspase-3, and apoptosis-induced cell death after PDT treatment, silicon phthalocyanine Pc 4, when compared with cells transfected with the control vector [76]. In contrast to the cell expressing wild-type SOD2, the Jurkat cells expressing mutated SOD2 were characterized by enhanced apoptosis-induced cell death after Pc 4 treatment, due to an increased generation of superoxide, activation of caspase-3, and apoptosis-induced cell death. These results were subsequently confirmed by comparing the mouse embryonic fibroblasts (MEFs) derived from SOD2-knockdown (Sod2 −/−) and wild-type (Sod2+/+) mice. Pc 4-based PDT, apoptosis-induced cell death was enhanced in Sod2 −/− MEFs through the induction of caspase-3-like activity, nuclear apoptotic changes, and ceramide accumulation compared with Sod2+/+ MEFs [76]. The intravenous administration of SOD2 following PDT treatment meaningfully reduced the curative effect of Photofrin-based PDT in SCCVII (squamous carcinoma cells) and RIF-1 and FsaR (fibrosarcoma cells) implanted in syngeneic C3H/HeN and BALB/c mice, respectively [77]. These results indicated that SOD could attenuate the cytotoxic effects of the superoxide radical generated by PDT, which promotes oxidative stress at the endothelium and NO/peroxynitrite-mediated apoptosis [77]. 

Soares’ group showed that SOD2 activity affected the cytotoxicity of PDT differently, depending on the main cytotoxic mediators of PDT [78]. Human lung adenocarcinoma A549 cells that exhibited lower SOD2 and catalase activities after PDT treatment were more sensitive to the cytotoxic superoxide anions and singlet oxygen produced by redaporfin-based PDT than mouse colon adenocarcinoma CT26 cells, but were resistant against the singlet oxygen produced after temoporfin-based PDT. In contrast to A549 cells, CT26 cells, which exhibited a higher activity of SOD2 and catalase after PDT treatment, were more resistant to superoxide radicals and singlet oxygen from redaporfin-based PDT, but were sensitive to the singlet oxygen from temoporfin-based PDT [78]. Consequently, SOD2 is one of the key enzymes in protecting tumor cells against cytotoxic superoxide anions mediated by PDT. Nonetheless, the SOD1 role against PDT was not prominent. Golab et al. showed that the addition of a cell-permeable SOD mimetic, MnTBAP, and the transient transgenic expression of SOD2 into urinary bladder cancer T24 cells significantly decreased cell susceptibility to the cytotoxicity of Photofrin-based PDT, compared with the control cells. On the other hand, T24 cells transfected only with the SOD1 expression plasmid did not show a modified cytotoxic effect from PDT [79]. In contrast, after knockdown of the SOD1 gene by siRNA in human cervical HeLa and oral squamous cell carcinoma Cal27 cells, an increase in singlet oxygen levels was obtained, which enhanced the cytotoxicity of zinc phthalocyanine-based PDT loaded into orthogonal upconversion nanoparticles (UCNPs). Furthermore, SOD1, siRNA-containing UCNPs significantly reduced tumor growth in Balb/c nude mice xenografted with Cal27 tumors, compared with the control [80].

### 2.2. Catalase 

Catalase is considered a crucial antioxidant enzyme in physiological conditions due to its catalyzing of the dismutation of H_2_O_2_ in a two-step reaction (Table 1). The first step is the oxidation of catalase, as follows: free catalase (ferricatalase; CAT-Fe^3+^) is oxidized by H_2_O_2_ into an intermediate, compound I (CAT-Fe^4+^O), and produces a water molecule (Figure 2). The second step is the reduction of compound I; compound I is reduced back to free catalase by another molecule of H_2_O_2_, producing one molecule each of water and oxygen. In total, catalase breaks down two molecules of H_2_O_2_ into two molecules of water and one molecule of oxygen [81,82]. Catalase is distributed throughout the body, particularly in the liver, kidneys, and erythrocytes. It sequentially protects healthy cells against ROS-mediated oxidative damage and tumor metastasis [83,84,85]. The importance of catalase has been demonstrated in numerous studies. For instance, catalase overexpression in mice mitochondria and heart extended their life span and reduced ischemia/reperfusion injury, respectively [86,87]. The administration of catalase conjugated with polyethylene glycol significantly inhibited hydrogen peroxide-induced lipid peroxidation in mice [88]. The gene mutation of catalase is related to diabetes, hypertension, vitiligo, or Alzheimer’s disease [89,90]. Moreover, hereditary catalase deficiency, acatalasemia, and hypocatalasemia, which were characterized by less than 10% and 50% of regular catalase activity, respectively, caused oral gangrene ulceration in Japanese or hypertension patients [91,92,93]. It is important to note that the role and activity of catalase undergoes enormous changes during tumor development [94,95].

To avoid the apoptosis-induced effects from HOCl and NO/peroxynitrite signaling, tumor cells establish cytoprotective systems consisting of membrane-associated catalase as a chief enzyme. Catalase abrogates HOCl and NO/peroxynitrite signaling through the dismutation of H_2_O_2_, the oxidation of NO, and the decomposition of peroxynitrite, respectively (Figure 5) [96,97,98,99]. This results in a decrease in HOCl and the hydroxyl radical production generated by HOCl and NO/peroxynitrite signaling, leading to attenuated apoptosis-inducing signaling in tumor cells. Interestingly, although extracellular singlet oxygen generated by PDT can inactivate a few catalase molecules of the cancer cell membrane and cause apoptosis, catalase still interrupts apoptosis-inducing signaling and diminishes PDT-induced cell death [100]. On the other hand, the inactivation of catalase may potentiate PDT-induced cell death, generating extracellular singlet oxygen. H_2_O_2_ dismutation, NO oxidation, and peroxynitrite decomposition no longer occur without catalase activity, leading to the formation of hydroperoxide radicals that can react with superoxide anions and generate singlet oxygen [101]. These reactions are specific in cancer cells due to a complex interaction between H_2_O_2_ and peroxynitrite [70].

Although the level of catalase expression varies among different types of cancer, the depletion of catalase expression or the inhibition of its enzymatic activity reduces tumor resistance to oxidative damage due to ROS-mediated therapies, including PDT [102,103]. One of the mechanisms exhibited by ROS-resistant glioma cells is the production of antioxidant enzymes, particularly catalase. Smith and co-workers reported that rat glioma-derived cell lines (C6, 36B10, RG2, and RT2), and human glioma cell lines (SNB19, U251, and A172) which overexpressed catalase protein showed elevated catalase enzymatic activity [104]. Knockdown of the *Cat* gene in 36B10 rat glioma cells significantly increased intracellular and extracellular ROS production, improving sensitivity against radiation (137Cs γ-irradiator) and H_2_O_2_ treatment [104]. Moreover, a study by Kang et al. elucidated that the synergistic action of the tumor suppressor p53 and the pro-apoptotic protein, p53-inducible gene 3 (PIG3), led to a diminution of catalase activity during UV-induced apoptosis. The resulting increase in ROS production enhanced the apoptotic cell death of HCT116, a human colon carcinoma cell line [105]. A study by Zhao and co-workers indicated that among three tested human tumor cell lines, a human hepatoma cell line, HepG2 cells, showed the highest antioxidant activity regulated primarily by catalase [106]. Whilst the glutathione redox cycle mainly controlled the other two cell lines, i.e., human cervical HeLa and lung adenocarcinoma A549 cells. The cytotoxicity of an ROS-dependent apoptosis inducer, artesunate, was affected by neither a γ-glutamylcysteine synthetase (GCS) inhibitor, L-buthionine-sulfoximine (BSO), nor glutathione (GSH) treatment in HepG2 cells. Whereas, a catalase inhibitor, 3-aminotriazole (3-AT), and catalase silencing meaningfully promoted sensitivity to H_2_O_2_- and artesunate-induced cytotoxicity. It was concluded that a strong resistance against ROS in HepG2 cells was dominated by catalase activity, while A549 and Hela cells exhibited a weak resistance against ROS due to a low catalase activity [106].

Klingelhoeffer and co-workers demonstrated that among 13 human cell lines, BT-20, a breast carcinoma, possessed the highest catalase protein level and activity, which led to cytotoxic resistance against the powerful pro-oxidant, H_2_O_2_, produced by ascorbic acid [103]. Notably, catalase-silenced BT-20 cells indicated a significant reduction in catalase protein level and its activity, which increased cell susceptibility against the ascorbic acid-mediated cytotoxic effect via elevating cell death and caspase-3/7 activity [103]. Glorieux et al. showed that human breast cancer cells, MCF-7, were sensitive to the combination of pro-oxidant drugs (ascorbate/menadione). Nevertheless, the elevation of the catalase protein level via treatment with 5-aza-2′-deoxycytidine, a DNA methyltransferase (DNMT) inhibitor, significantly developed a cell resistance phenomenon [102]. A decrease in catalase activity via treatment with 3-AT, a catalase inhibitor, meaningfully improved MCF-7 cell sensitivity to the combination of pro-oxidant drugs. Notably, the catalase overexpression in MCF-7 cells did not result in cell resistance against conventional chemotherapies such as cisplatin, 5-fluorouracil, doxorubicin, or paclitaxel. It suggested that altering antioxidant enzyme expression, including catalase, might cause a resistance mechanism only towards redox-based chemotherapeutic agents in tumor cells [107]. Among 32 patients suffering from mesothelioma, 24 (75%) cases indicated catalase expression in tumor cells. Additionally, a human mesothelioma cell line, M38K, showed a high catalase protein level and the catalase inhibition by 3-AT improved cells’ sensitivity to epirubicin [108]. In the case of PDT, the exogenous addition of catalase potentially reduced the LD_50_ of hematoporphyrin-based PDT by 60% in Ehrlich ascites carcinoma (EAC) cells, suggesting that an increase in catalase can promote cell resistance against PDT-induced cytotoxicity [109].

### 2.3. Glutathione Redox Cycle

Not only is catalase responsible for removing H_2_O_2_, but is also responsible for the removal of nicotinamide adenine dinucleotide phosphate (NADPH)-dependent antioxidants; glutathione (GSH; L-γ-glutamyl-L-cysteinyl-glycine) and thioredoxin-peroxiredoxin redox systems facilitate elimination of H_2_O_2_ (Figure 2) [110,111]. GSH is an intracellular antioxidant molecule that plays a crucial role in maintaining redox status, xenobiotic metabolism, and regulation of gene expression and programmed cell death (Table 1). Two ATP-dependent enzymes regulate the synthesis of GSH, as follows: (1) glutamate–cysteine ligase (GCL) or γ-glutamylcysteine synthetase (GCS) that facilitates the formation of γ-glutamyl cysteine using glutamic acid and cysteine, and (2) GSH synthase (GS) that catalyzes the addition of glycine to the dipeptide (Figure 6). The γ-glutamyl bond makes glutathione more resistant to degradation by cellular proteases, which typically cleave α-peptide bonds. Additionally, the presence of the γ-glutamyl bond has implications for transporting glutathione and its precursor amino acids across cell membranes. Specific transporters in cell membranes recognize the γ-glutamyl moiety and facilitate the uptake of glutathione [112]. Under oxidizing conditions, GSH shows antioxidant activity via acting as a reducing agent or electron donor to reduce H_2_O_2_, lipid hydroperoxides (LOOHs), and peroxynitrite, before being converted to oxidized glutathione or glutathione disulfide (GSSG) by glutathione peroxidase (GPx), resulting in a decrease in the GSH/GSSG ratio. GSSG can be converted back to GSH using NADPH, catalyzed by glutathione reductase (GR) [113,114,115].

Furthermore, glutathione *S*-transferase (GST), which is considered a xenobiotic-conjugating enzyme in phase II metabolism, can further maintain the cellular redox state and defense against cytotoxic ROS through a wide range of molecular mechanisms, such as the Jun N-terminal kinase, apoptosis signaling kinase 1, 4-hydroxy-2-transnonenal, and mitogen-activated protein kinase pathway [116,117,118,119,120,121].

The glutathione redox cycle is a crucial cytoprotective mechanism against cytotoxic ROS in tumor cells. Zhao et al. showed that the glutathione redox cycle potentially affected ROS-mediated anticancer therapies, H_2_O_2_, and artesunate-induced apoptosis in HeLa and A549 cells [106]. Treatment with a GCS inhibitor, buthionine sulfoximine (BSO) eliminated the cell resistance to H_2_O_2_ and enhanced the H_2_O_2_-induced cytotoxicity in HeLa and A549 cells, which were regulated by GPx and catalase activity. Moreover, the pretreatment of the cells with GSH abolished the cytotoxicity of artesunate. It is important to note that artesunate cytotoxicity was not affected by BSO or GSH treatment in HepG2 cells, which was mainly controlled by catalase [106]. The glutathione redox cycle plays an essential role in ROS cytotoxicity generated by PDT. Lee et al. showed a crucial role of GSH in the sensitivity of chlorin e6 (Ce6)-based PDT in two human cholangiocarcinoma cell lines, HuCC-T1 and SNU1196 [122]. SNU1196 cells possessed higher GSH basal levels, catalytic subunit GSC expression, and GPx and GR activity compared with HuCC-T1 cells. Thus, HuCC-T1 cells were more sensitive to Ce6-based PDT, exhibiting three and two times higher apoptotic and necrotic cell deaths, respectively, than SNU1196 cells. The addition of exogenous GSH to HuCC-T1 cells reduced the ROS generation and cytotoxicity of Ce6-based PDT. At the same time, a GPx inhibitor, mercaptosuccinic acid (MSA), displayed opposite effects in SNU1196 cells [122]. After the addition of an exogenous GSH, GSH monoethyl ester significantly abrogated ROS induction and apoptosis mediated by hypericin-based PDT in human colorectal cancer HCT8 and HCT116 cells [123]. Wang and co-workers reported that GPx protected MCF-7 cells against lipid peroxidation, leading to cell resistance against singlet oxygen-induced oxidative damage due to Photofrin-based PDT. GPx-overexpressed MCF-7 cells indicated a lower level of a lipid peroxidation marker, LOOH, suggesting that GPx could repair lipid peroxidation. GPx minimized membrane damage and singlet oxygen-induced apoptosis, increasing cell survival after PDT treatment [124]. Likewise, Dabrowski et al. reported that GST facilitated the resistance to PDT. The human kidney fibroblast K293 cell line was transfected with a plasmid-encoding green fluorescent protein and GSTP1-1 (pIRES-GFP-GSTP) to increase the expression of GSTP1-1. The GSTP1-1-overexpressed K293 cells showed resistance against PDT by reducing the hypericin-based PDT cytotoxicity [117]. Furthermore, in 2004, Lu and Atkins reported that the production of ROS induced by hypericin can be attenuated by the ligandin of GST [125]. Interestingly, hydrogen sulfide (H_2_S) has been observed to diminish the activity of PDT-based therapy via ROS/RNS scavenging [126]. The cytotoxicity of 5-ALA was dramatically reduced following exposure to H_2_S in the murine adenocarcinoma LM2 cell line. The outcomes were associated with an elevation of GSH levels and catalase activity, with a reduction in singlet oxygen level [127].

### 2.4. Heme Oxygenase-1 (HO-1)

Heme oxygenase (HO) is known as an enzyme that metabolizes pro-oxidant heme into the antioxidant biliverdin (which is converted to bilirubin), iron (Fe^2+^), and carbon monoxide (CO) (Table 1) [128,129,130]. The heme catabolism pathway is presented in Figure 7. There are three isoforms of HO, HO-1, HO-2, and HO-3, encoded by separate genes [131,132,133]. Genes encode heme oxygenase-1 (*Hmox1*) and heme oxygenase-2 (*Hmox2*) that are located on chromosome 22 and 16, respectively [134,135,136]. HO-1 protein (32.8 kDa) is highly expressed in spleen, liver, bone marrow, and senescent red blood cells [134,137,138], while HO-2 protein (36 kDa) is expressed in brain, testis, and endothelial and smooth muscle cells from cerebral vessels [139,140,141]. HO-3 is thought to be a pseudogene derived from HO-2 transcripts and its function remains unknown [133].

HO-1 is known to have essential functions in heme catabolism. Moreover, it has anti-inflammatory, anti-apoptotic, and cytoprotective effects [142,143,144,145,146]. Its upregulation has been observed in response to various stressors, including oxidative stress, which may be induced by PDT [142,143,144,145,146]. HO-1 expression is also regulated by several transcription factors, such as Nrf2 (nuclear factor erythroid 2-related factor 2), AP-1 (activator protein-1), Bach1, and HIF (hypoxia-inducible factor) [143,144,147,148].

The degradation products of heme are essential biologically active compounds (Figure 7). Carbon monoxide (CO) is a toxic gas that can be deadly in high concentrations [149]. However, some studies have shown that low levels of CO generate cell-protective effects. Moreover, CO has anti-inflammatory and anti-apoptotic effects, which can protect cells from the damage caused by oxidative stress [150,151,152,153]. Biliverdin is a green pigment produced from the breakdown of heme via OH-1. Biliverdin is then converted into bilirubin via the enzyme biliverdin reductase. Biliverdin and bilirubin are powerful antioxidants that can scavenge peroxyl radicals and prevent lipid peroxidation. Bilirubin demonstrates scavenging abilities against superoxide, peroxyl radicals, and peroxynitrite [154,155,156,157,158].

The role of HO-1 in PDT resistance has been investigated in studies conducted on various cancer cell lines. Cytotoxicity analysis confirmed that HO-1 exhibits a cytoprotective effect on the cells attacked by ROS-based therapy, including PDT [142,143,144,145]. It was demonstrated that HO-1 protects tumor cells against PDT-mediated cytotoxicity by alleviating oxidative stress and promoting cell survival [142]. Furthermore, the induction of HO-1 in rat malignant meningioma cells subjected to talaporfin sodium-mediated PDT was explored, emphasizing the upregulation of HO-1 as a cellular response to oxidative stress induced by PDT [143,144]. Also, these findings were extended by highlighting the influence of HO-1 on the outcome of 5-aminolevulinic acid (5-ALA)-mediated PDT in melanoma cells [145]. Collectively, these investigations underscore the significance of HO-1 as a key player in the cellular defense against PDT-induced oxidative stress, providing valuable insights into the mechanisms underlying the cytoprotective effects observed in cells exposed to photodynamic treatment.

In addition, PDT induces the expression of *Hmox1* [142,143,144,146], which may contribute to the acquisition of resistance to PDT by cancer cells [143,144]. Several studies also suggest that inhibiting HO-1 activity or expression can sensitize cancer cells to PDT and enhance the therapeutic outcome of this treatment. Several studies have explored the use of HO-1 inhibitors to increase the effectiveness of PDT [142,145,159,160]. Cheng et al. reported that a natural antioxidant, alpha-lipoic acid (ALA), can increase cell resistance against oxidative stress. ALA administration significantly increased the expression and activity of HO-1 in rat aortic smooth muscle A10 cells, further reducing cytotoxicity after H_2_O_2_ treatment. Interestingly, the cytotoxicity of H_2_O_2_ can be reversed by treatment with an HO-1 inhibitor, Zn(II) protoporphyrin IX (ZnPPIX) [161]. In the study of the role of mitogen-activated protein kinase (p38MAPK) and phosphatidylinositol 3-kinase (PI3K) pathways in HO-1 induction, it was found that the application of p38MAPK and PI3K inhibitors, PD169316 and LY294002, respectively, significantly reduced HO-1 expression [146]. Inhibitor application enhanced the cytotoxicity of hypericin-based PDT in human urinary bladder carcinoma T24 cells. In addition, the transfection of T24 cells with an HO-1 siRNA significantly promoted apoptosis and increased the caspase-3 level after hypericin-based PDT treatment. It indicates that a reduction in HO-1 expression can enhance susceptibility to apoptosis-mediated PDT [146].

Activity of HO-1 in PDT can be reduced by using pharmacological agents, namely well-known and characterized metalloporphyrins (MPs) [159,162,163,164,165]. Metalloporphyrins are porphyrins in which the porphyrin macrocycle is chelated to a metal cation, acting as a tetradentate ligand [162]. MPs do not have oxygen-binding capacity and they are not subject to oxidative degradation. MPs also exhibit significantly greater binding affinity to HO isoforms compared to heme [163]. There are several studies which have confirmed that MPs cause a decrease in HO-1 activity. Nowis et al. showed that ZnPPIX increased the cytotoxic effect of PDT in colon and ovarian carcinoma cells [142]. Also ZnPPIX-mediated HO-1 inhibition increased the sensitivity of nasopharyngeal carcinoma cells to radiotherapy [164]. There are also studies showing that another MP, SnPPIX, enhanced the effect of PDT on melanoma tumors and reduced the growth of Kaposi sarcoma in vivo [159,165].

Overall, these findings suggest that HO-1 plays a critical role in PDT resistance and targeting this enzyme could be a promising approach to improve the efficacy of PDT in cancer treatment. However, further research is needed to determine the optimal strategy for targeting HO-1 in combination with PDT.

**Table 1 ijms-25-03164-t001:** Antioxidant enzymes in cancer treatment.

Antioxidant Enzymes	Target of Action	Location	Function	Therapeutic Effects on Cancers	Ref.
SOD	Superoxide anion	SOD1: Cytoplasmic compartments SOD2: Mitochondria SOD3: Extracellular space	Dismutation of superoxide anion	Anticancer agents Tumor-growth suppressor Apoptosis induction Chemo- and radioprotectors Protection of normal cells against ionizing irradiation and chemotherapeutic agents SOD mimetics Induction of SOD activity by mimetic molecules Enhancement of anticancer agent activities in combination therapy Protection of normal cells against chemotherapeutic agents Therapeutic targeting SOD SOD2 promotes resistance to apoptotic cell death of cancer cells SOD2 facilitates cancer cell growth	[42,56,166,167,168,169]
Catalase	H_2_O_2_	Peroxisomes	Dismutation of H_2_O_2_	Catalase mimetics Induction of catalase activity by mimetic molecules Enhancement of anticancer agent activities in combination therapy Therapeutic targeting catalase Catalase prevents oxidative injury and apoptotic cell death of cancer cells	[85,95,169]
GCS	Glutamic acid and cysteine	Cytosol	Synthesis of glutathione	Anticancer agents Enhancement of anticancer agent activities in combination therapy	[170,171]
GR	GSSG	Cytosol	Conversion of GSSG to GSH	Therapeutic targeting GR The accumulation of GSSG in the cell can exert a pro-oxidative effect, Inhibiting its reduction may enhance the efficacy of some anticancer drugs.	[170]
GPx	H_2_O_2_	Cytosol and mitochondria	Reduction of H_2_O_2_	Therapeutic targeting GPx Target of chemotherapy, hemodynamic therapy, and photodynamic therapy	[172]
GST	Xenobiotics	Microsome, mitochondria, and cytosol	Conjugation of xenobiotics with GSH	Therapeutic targeting GST GSH conjugation with anticancer drugs catalyzed by GST protects cancer cells. GST isoenzymes have become promising targets for therapy due to their overexpression in a wide range of tumors.	[115,120]
HO-1	Heme	Endoplasmic reticulum, mitochondria, the vacuole, nucleus, and plasma membrane	Metabolism of heme into biliverdin	Predictive marker HO-1 expression is associated with cancer response rate	[130,173]

Note. GCS, γ-glutamylcysteine synthetase; GPx, glutathione peroxidase; GR, glutathione reductase; GSH, glutathione; GST, glutathione *S*-transferase; GSSG, oxidized glutathione; HO-1, heme oxygenase-1; SOD, superoxide dismutase.

## 3. The Inhibitors of Antioxidant Enzymes Used to Overcome Cancer Resistance to PDT

Alteration of the level of the antioxidant molecules in cancer cells is one of the essential cytoprotective mechanisms that occur during PDT. Below, we summarize the strategies based on the application of inhibitors against the antioxidant enzymes that were studied, in order to improve the effectiveness of PDT. 

### 3.1. SODs Inhibitors

#### 3.1.1 2-Methoxyestradiol (2-ME, SOD2 Inhibitor)

2-Methoxyestradiol, an estrogen metabolite, has been confirmed to be an enzymatic inhibitor of human and bovine SOD1 and *E.coli* SOD2 (Figure 8) [79,174]. Nonetheless, 2-ME was used as a selective SOD2 inhibitor in the study of antioxidant enzyme inhibitors against PDT [78,175]. 2-ME can enhance PDT-induced cell death in various human tumors. Corresponding to the report of Golab and co-workers, 2-ME showed a synergistic effect and enhanced the cytotoxic effect of Photofrin-based PDT in a dose-dependent manner in three murine cell lines (i.e., C-26, LLC, and MDC), five human cell lines (i.e., T47-D, PANC-1, HPAF-II, HPAC, and T24), and two xenograft tumor mice models (i.e., C-26 and LLC). Interestingly, C-26 adenocarcinoma-implanted Balb/c mice were cured by 60% after 14 days of 2-ME and Photofrin-based PDT treatment [79]. In the study of the cell cytoskeleton in human ovarian clear carcinoma OvBh-1 and breast adenocarcinoma MCF-7 cells, the 2-ME application, together with PDT, immediately induced cell shrinkage, disruption of actin filaments and microtubules architecture, followed by the reorganization of the cytoskeleton and nucleus [176]. 2-ME enhances PDT-induced apoptosis through an increase in caspase 3/7 and 12, with a lowering ratio of anti-apoptotic protein (Bcl-2) to pro-apoptotic protein (Bax). Kimáková et al. showed that SOD activity and its mRNA were significantly increased by hypericin-based PDT in breast adenocarcinoma MCF-7 cells, while this effect was not observed in MDA-MB-231 cells [177]. Treatment with 2-ME can prevent SOD mRNA expression using hypericin-based PDT, which further reduced the proliferative capability of MCF-7 cells with an increase in the caspase 3/7 activity and an enrichment of nucleosomes (DNA fragmentation). The 2-ME and PDT co-treatment further lowered the Bcl-2/Bax ratio from 0.19 to 0.5, compared with a single treatment of cells using hypericin-based PDT. It indicated that 2-ME can potentiate caspase 3/7 and apoptosis induction using hypericin-based PDT [177]. The enhancement of apoptosis due to 2-ME and cyanine IR-775-based PDT was observed in human breast (MDA-MB-231) and ovarian (SKOV-3) adenocarcinoma cells. Co-treatment with 2-ME and IR-775 significantly decreased MDA-MB-231 and SKOV-3 cell viability after 24 and 72 h, compared with treatment with 2-ME or IR-775 alone. Furthermore, the expression of SOD2 and caspase 12 were increased after cells’ co-treatment with 2-ME and IR-775 in both cell lines [178].

#### 3.1.2. Diethyldithiocarbamate (DDC, SOD1 Inhibitor)

Diethyldithiocarbamate, the primary metabolite of disulfiram, is a hydrophilic metal-chelating agent for Cu(II) and is well known as a SOD1 inhibitor (Figure 8) [175,179,180]. It is important to note that DDC can also decrease GSH levels [181,182]. DDC was investigated as a cytotoxicity enhancer for several photosensitizers [109]. The addition of DDC to zinc phthalocyanine-based PDT significantly enhanced the cytotoxicity of both free and liposome-encapsulated zinc phthalocyanine in mouse embryo fibroblast NIH3T3 and human breast cancer MDA-MB-231 cells, compared with zinc phthalocyanine treatment alone. The IC_50_ of zinc phthalocyanine-based PDT decreased from 64.9 to 32.43 and 6.78 to 3.39 μg/mL after co-treatment with DDC on mouse embryo fibroblast (NIH3T3) and MDA-MB 231 cell models, respectively. Co-encapsulation into liposomes of DDC and zinc phthalocyanine was confirmed to reduce SOD and GSH activities in MDA-MB 231 cells that resulted in sensitizing tumor cells to PDT [183]. The enhancement effect of DDC was also indicated in an in vivo study, in which DDC significantly potentiated an ear swelling response after Photofrin II-based PDT in female C3H mice, while the control group did not show any significant response [184]. Nevertheless, the enhancement effect of DDC in PDT was not shown in some tumors. Wright et al. investigated the effect of three antioxidant enzyme inhibitors delivered together with meta-tetrahydroxyphenyl chlorin (mTHPC)-based PDT in a murine dorsal root ganglion model composed of their neuron and associated satellite glia cells. The pretreatment of DDC at a dose of 50 μM significantly increased the sensitivity of neuron cells only, and an SOD2 inhibitor, 2-ME, at a dose of 1 μM, did not show any significant enhancing effects, whilst BSO pretreatment at a dose of 500 μM affected both neurons and satellite glia cells. Interestingly, a combinatory pretreatment of cells with DDC and BSO exhibited a near total loss of neuron and satellite glia cells after mTHPC-based PDT [185].

### 3.2. Catalase Inhibitors

#### 3-Aminotriazole (3-AT, Catalase Inhibitor)

3-aminotriazole is a triazole heterocycle derivative exhibiting several effects, including histamine H_2_-receptor antagonistic, anti-asthmatic, and herbicidal (Figure 8) [186,187,188]. 3-AT is further used as a specific catalase inhibitor that can irreversibly inhibit catalase activity in the presence of a low and constant H_2_O_2_ level [189,190]. Many studies reported that concomitant treatment of 3-AT and PDT significantly improved PDT-induced cell death in various tumor cells [191,192]. Price and co-workers demonstrated the pro-apoptotic effect of H_2_O_2_ in benzoporphyrin derivative-based PDT [193]. Apoptosis-mediated PDT was enhanced due to 3-AT treatment by reducing the LD_50_ by 23.0% in a model murine leukemia P388 cell line. At the same time, the application of a catalase enzyme analogue, CAT^-SKL^, and a ferrous iron chelating agent, 2,2-bipyridyl (BID), showed the opposite effect, protecting the cells from photokilling by increasing the LD_50_ of 18.0 and 8.2%, respectively. The conversion of procaspase-3 to active caspase-3 and the increase in H_2_O_2_ production were also confirmed in the cells after PDT treatment with 3-AT [193]. Soares’ group reported that 3-AT can significantly enhance the cytotoxicity of redaporfin-based PDT in the low SOD and catalase activity cell line, A549. Whilst this effect did not occur in CT26 cells which showed higher SOD and catalase activity after PDT treatment [78]. Moreover, the effects of 3-AT were displayed in animal models. Catalase activity was significantly reduced by 3-AT pretreatment in EAC implanted mongrel mice. Pretreatment of 3-AT further enhanced the cytotoxicity of hematoporphyrin-based PDT by decreasing the LD_50_ of the PDT. The authors concluded that catalase potentially protects tumor cells against PDT [109]. The combination of Photofrin II-based PDT with a catalase inhibitor, 3-AT or hydroxyl amine, or with an SOD inhibitor, DDC, meaningfully promoted the potentiation of ear swelling response in female C3H mice. While the control group did not show any significant response. The authors suggested that superoxide anion and H_2_O_2_ were involved in Photofrin II-mediated cutaneous photosensitization [184].

### 3.3. Inhibitors Involved in Glutathione-Related Enzyme Systems

#### 3.3.1. L-Buthionine Sulfoximine (BSO, GCS Inhibitor)

Guthionine sulfoximine was synthesized in 1979 as a specific enzymatic inhibitor of γ-glutamylcysteine synthetase (GCS) in the glutathione synthesis pathway (Figure 8) [194]. BSO caused GSH deficiency, sensitizing tumor cells to PDT-induced cell death [109,185,195]. Treatment with 0.002–10 mM BSO significantly depleted cellular GSH level in a dose-dependent manner in Chinese hamster ovary (CHO), Chinese hamster lung V-79, mouse-derived breast cancer EMT-6, and fibrosarcoma RIF cell lines. BSO further enhanced the cytotoxicity of Photofrin II-based PDT in all cell lines [196]. Likewise, the cytotoxicity of hematoporphyrin was enhanced when co-administered with BSO in murine leukemia L1210 cells [197]. BSO was employed in the microsphere system together with PDT agents to increase the cytotoxicity. BSO potentiated PDT cytotoxicity, mediated by Ce6 conjugated with an ethyldimethyl aminopropylcarbon diamide-activated polystyrene microsphere, in human bladder carcinoma MGH-U1 cells. It is important to note that the BSO’s potentiation effect was not indicated after applying the Ce6-based PDT without conjugation to the microspheres [198]. In the study of the pro-oxidant and antioxidant effect of ascorbate performed by Soares et al., the inhibiting effects of 2-ME, 3-AT, and BSO were investigated [78]. 2-MT, 3-AT, and BSO significantly promoted the cytotoxic effect of superoxide anions and singlet oxygen produced by redaporfin-mediated PDT in A549 cells. Moreover, the application of the pro-oxidant molecule, ascorbate, can even further enhance this effect. Conversely, these inhibitors did not affect CT26 cells due to their high SOD2 activity and up-regulation of catalase after PDT treatment. In contrast to redaporfin, temoporfin that mainly generated singlet oxygen showed stronger cytotoxicity in CT26 than in A549 cells. It was suggested that the PS generated both superoxide anions and singlet oxygen, which offered better opportunities for cytotoxicity mediated by PDT in combination with antioxidant enzyme inhibitors and/or ascorbate. Interestingly, in the combination of PDT and ascorbate, ascorbate did not act only as a pro-oxidant molecule enhancing cytotoxic effects of superoxide anions and singlet oxygen, but also acted as an antioxidant, which quenched singlet oxygen, influencing the redox status of the cells [78]. However, the enhancement of the cytotoxic effect by BSO was not indicated after treatment of hypericin-based PDT in GPx4-expressing MCF-7 cells. Theodossiou and co-workers reported the cytotoxic effect of hypericin-based PDT in two phenotypically and genotypically different breast adenocarcinoma cell lines, MCF-7 and MDA-MB-231 [199]. High GSTP1-expressing MDA-MB-231 cells were more sensitive to hypericin than GPx4-expressing MCF-7 cells. Though co-treatment of BSO and hypericin-based PDT caused total GSH depletion in MCF-7 and MDA-MB-231 cells, the enhancement of the PDT cytotoxicity was exhibited only in MDA-MB-231 cells. Nevertheless, PDT cytotoxicity of MCF-7 cells can be enhanced in a combinatory treatment with BSO and BCNU, a GR inhibitor. The authors concluded that the GPx-4 enzyme could be used as a predictive marker for cell response to PDT, whilst GST corresponds to the chemoresistance of a cell line [199]. Kimani’s group reported cytotoxicity enhancement after the application of four antioxidant enzyme inhibitors in PDT employing disulphonated aluminum phthalocyanine (AlPcS_2_) as a photosensitizer [175]. BSO significantly reduced MCF-7 cell viability after 24 h of incubation with AlPcS_2_, compared with AlPcS_2_ treatment alone. Whilst cells’ pretreatment with 2-ME, DDC, or 3-AT and AlPcS_2_-based PDT did not result in any effect. The cytotoxicity of AlPcS_2_-based PDT was further enhanced after a 24 hr preincubation with BSO plus 3-AT or 2-ME, or with a combination of four antioxidant enzyme inhibitors. The authors concluded that the most related antioxidant mechanism in protecting MCF-7 cells against PDT is a glutathione redox cycle, followed by SOD2, catalase, and SOD1 [175]. Lee et al. showed that BSO can enhance the effectiveness of PDT in tumors characterized by either a high or low GSH level [200]. Among ten tumor cell lines, colorectal HCT116 and ampulla vater carcinoma SNU478 cells exhibited the highest and lowest GSH level, respectively. The combination of BSO and Ce6-based PDT significantly reduced the total GSH level and increased ROS generation in both HCT116 and SNU478 cells, compared with Ce6 treatment alone. The addition of intracellular GSH levels using glutathione reduced ethyl ester displayed a cytoprotective effect against PDT, by providing higher HCT116 and SNU478 cell survival after co-treatment with BSO and Ce6-based PDT, compared with non-treated cells. Notably, BSO enhanced the cytotoxicity for a wider concentration range of Ce6 treatment in HCT116 than SNU478 cells. It suggests that BSO induced a less synergistic effect with Ce6-based PDT in cells that have a lower GSH level [200]. BSO with Photofrin significantly decreased the cell survival, colony-forming capacity, and invasion properties of two human glioma cell lines, U87 and U251n, in a dose-dependent manner, compared with Photofrin treatment alone. Therapy enhancement using BSO in combination with Photofrin was also confirmed in a xenograft rat model. Co-treatment of BSO with Photofrin-based PDT showed remarkable superficial tumor damage, necrosis, and focal hemorrhage in the brain of U87 glioma-implanted rats, while the tumor lesion after treatment with only Photofrin was not obvious. The lesion volume of xenografted U87 glioma in the rat brain after BSO and Photofrin treatment was meaningfully greater [201]. Moreover, co-treatment with BSO and Photofrin-based PDT significantly increased the lesion volume of tumor necrosis in the brain of male Fischer rats bearing an intracerebral 9L gliosarcoma, compared with Photofrin treatment alone. Interestingly, BSO did not increase lesion volume in the normal brain samples [202]. Additionally, both in vitro and in vivo studies of BSO application in nanospheres were performed. Chlorin e6-loaded poly(ethylene glycol)-block-poly(D,L lactide) nanoparticles (Ce6-PEG-PLA-NPs)-based PDT exhibited synergetic effects with BSO. The co-administration of BSO and Ce6-PEG-PLA-NPs significantly decreased cell viability compared with Ce6-PEG-PLA-NPs treatment alone in mouse squamous cell carcinoma SCC-7 cells. The results were confirmed in the SCC-7 xenografted mouse model. The co-treatment completely suppressed tumor growth after 14 days, while the treatment of Ce6-PEG-PLA-NPs alone showed a lower effect. Hematoxylin and eosin-staining showed increased apoptosis and tissue damage after the combinatory therapy. Additionally, the body weight and liver and kidney tissues of treated mice did not significantly change due to therapy, referring to a negligible systemic toxicity of the NPs and BSO [203]. Similarly, BSO co-encapsulated with indocyanine green in thermosensitive liposomes was employed [204]. This delivery system was named as an IR photothermal liposomal nanoantagonist (PLNA), due to it containing BSO, a GSH biosynthesis antagonist. The presence of BSO in this delivery system resulted in a significant depletion of intracellular GSH levels in murine breast cancer 4T1 cells, when compared to nanocarriers lacking BSO. The results revealed that PLNA application increased the intracellular ROS levels, PDT cytotoxicity, and the apoptosis rate of the cells. Furthermore, PLNA demonstrated interesting outcomes in murine breast cancer 4T1-implanted BALB/c mice. Notably, PLNA not only significantly reduced GSH levels, tumor growth, and weight after 16 days of treatment, but also exhibited in vivo biosafety. After 30 days of treatment, alterations were not evidenced in the histological morphology of vital organs, levels of biochemical indicators of liver and kidney function, and essential blood biochemistry parameters [204].

In addition to GSH depletion by BSO treatment, the inhibition of deubiquitinating enzymes substantially interferes with cancer cell survival upon oxidative stress [205]. Harris et al. indicated that the inhibition of deubiquitinating enzymes in combination with BSO treatment increased cell death by ferroptosis, a cell death pathway activated by excessive lipid peroxidation [206]. Therefore, the cytotoxicity of PDT-based therapy is feasibly enhanced by deubiquitinating enzyme inhibitors combined with BSO.

#### 3.3.2. 1,3-Bis(2-chloroethyl)-1-nitrosourea or Carmustine (BCNU, GR Inhibitor)

Not only an alkylating agent, 1,3-bis(2-chloroethyl)-1-nitrosourea or carmustine is extensively used in the clinical treatment of malignant gliomas, but also as a selective inhibitor of GR in the glutathione redox cycle, which can further enhance PDT-induced cell death in several conditions (Figure 8) [207,208,209,210]. The study of fractionated light delivery in PDT found that BCNU increased ROS levels by diminishing the GSH detoxification that enhanced the PDT cytotoxicity. Although fractionated irradiation of aluminum (III) phthalocyanine tetrasulfonate (AlPcS_4_)- and hypericin-based PDT decreased ROS production and PDT cytotoxicity in human epidermoid carcinoma A431 cells, BCNU could reverse these effects by enhancing the ROS production and cytotoxicity of both PDT types [211]. In the study of glucose deprivation on PDT efficacy, two antioxidant inhibitors, BCNU and BSO, significantly increased intracellular ROS generation and enhanced the cytotoxicity of AlPcS_4_-based PDT, compared with AlPcS_4_ alone, in human epidermoid carcinoma A431 cells. Moreover, co-treatment of AlPcS_4_ with BCNU or BSO exhibited PDT-induced apoptosis via an increase in the caspase-3-like enzyme activity and nuclear fragmentation. It is important to note that BSO was more potent than BCNU, since BCNU did not decrease intracellular GSH levels, while BSO inhibited enzymes in approximately 90% [195]. Sun and co-workers showed that BCNU sensitized SOD2-overexpressing cells to Photofrin-based PDT [212]. Human breast carcinoma ZR-75-1 cells were transfected with an adenoviral construct containing the cDNA for *SOD2* (Ad*MnSOD*) to increase SOD2 expression. The cytotoxicity of Photofrin-based PDT did not change in ZR-75-1 cells together with Ad*MnSOD*, compared with ZR-75-15 cells alone, though a combination of Ad*MnSOD* with BCNU could significantly increase ROS accumulation and further enhanced the cytotoxicity of Photofrin. Authors hypothesized that an increase in the conversion of superoxide anions to H_2_O_2_ by elevating SOD2 levels with simultaneous inhibition of GR by BCNU could enhance the steady-state levels of superoxide and tumor cell killing [212]. BCNU enhanced PDT-induced cell death in a PDT-resistant cell line, MCF-7 cells, which highly expressed GPx4. Although BSO showed only a slight enhancing effect of the hypericin-based PDT in MCF-7 cells, a combination of BCNU and BSO significantly enhanced the cytotoxicity of the PDT [199]. In the study of the temperature effect on hematoporphyrin-based PDT efficacy, the pretreatment of BCNU in EAC-implanted mongrel mice remarkably reduced the GR activity and enhanced PDT-induced cell death via reducing the LD_50_ of the PDT. Likewise, pretreatment of the other three antioxidant enzyme inhibitors, DDC, BSO, and 3-AT, reduced the activity of antioxidant enzymes and lowered the LD_50_ of the PDT significantly [109,213]. Interestingly, these inhibitors could not only enhance PDT-induced cell death via reducing the activities of antioxidant enzymes, but also suppressed the photoinduced degradation of HPD [214].

#### 3.3.3. Mercaptosuccinic Acid (MSA, GPx1 Inhibitor)

Mercaptosuccinic acid is a mercaptan derivative, which is generally employed as a GPx1 inhibitor due to competing with GSH to bind to the active site, active-site selenocysteine (Figure 8) [215,216]. Due to diminishing GPx activity, which is an antioxidant enzyme in the glutathione redox cycle, MSA could enhance PDT-induced cell death [217]. The study on the effectiveness of Rose Bengal-based PDT by Yao et al. indicated that primary human skin fibroblasts (FBs) grown in monolayers were more sensitive to the therapy than fibroblasts grown in collagen gels. However, the cytotoxicity of Rose Bengal-based PDT in fibroblasts grown in collagen gels was enhanced when combined with either an antioxidant enzyme inhibitor, MSA or 3-AT [192]. Lee and co-workers demonstrated the effect of combined treatment with MSA and Ce6-based PDT on cholangiocarcinoma cells. The authors used two different cell lines, intrahepatic HuCC-T1 and extrahepatic SNU1196. Both lines differed in GSH level, with SNU1196 cells having higher GSH basal levels, catalytic subunit GSC expression, and GPx and GR activity compared with HuCC-T1 cells. Nevertheless, co-treatment with MSA and Ce6 significantly decreased GPx activity, resulting in elevating ROS levels and cytotoxicity in SNU1196 cells [122].

#### 3.3.4. 9-Chloro-6-ethyl-6H[1,2,3,4,5]pentathiepino[6,7-b]indole (CEPI, GPx1 Inhibitor)

9-chloro-6-ethyl-6H[1,2,3,4,5]pentathiepino[6,7-b]indole is a pentathiepin derivative that was recently developed as a potent GPx1 inhibitor (Figure 8). Among eight pentathiepin derivatives, CEPI showed the strongest inhibitory activity against bovine erythrocyte GPx activity, which was 15-fold higher than MSA [218]. Lange and Bednarski reported synergistic effects of mTHPC-based PDT with a GPx1 inhibitor, MSA or CEPI, in a few cell lines. Among the five human cell lines, co-treatment with CEPI and mTHPC showed a synergistic effect in esophageal carcinoma KYSE-70 and urinary bladder carcinoma RT-4 cells via increased cytotoxicity, ROS generation, and apoptosis. MSA synergized with mTHPC-based PDT in lung carcinoma A-427, oral carcinoma BHY, KYSE-70, and RT-4 cells. It is important to note that MSA or CEPI could also produce additive or antagonistic effects against mTHPC-based PDT depending on doses and types of tumor [217].

#### 3.3.5. GST Inhibitors

##### Ethacrynic Acid (ECA, GSTP1-1 Inhibitor)

Ethacrynic acid is an FDA-approved GSTP1-1 inhibitor, which has been used as a diuretic and alkylating agent in cancer treatment that can reverse anticancer resistance by inhibiting GST activity (Figure 8) [219,220,221]. In addition, ECA has been extensively used as an inhibitor of GST in order to enhance the effectiveness of several tumor treatments, including PDT [222,223,224]. Won et al. showed new constructs of brominated BODIPY-based PDT conjugated with ethacrynic acid (EA-BPS) [225]. Not only does EA-BPS exhibit more potent cytotoxicity compared with free brominated BODIPY-based PDT in human breast adenocarcinoma MDA-MB-231 and MCF-7 cells, but it also showed higher ROS production and formation of singlet oxygen, superoxide anions, hydroxyl radicals, and peroxynitrite anions in MCF-7 cells. Furthermore, EA-BPS significantly reduced tumor volume in MDA-MB-231-implanted immunodeficient nude mice without alteration of body weights or levels of aspartate transaminase (AST), alanine aminotransferase (ALT), and creatinine activity [225].

##### SX-324 (GSTP1-1 Inhibitor)

A novel GSTP1-1 inhibitor, SX-324, was developed based on a previous GSTP1-1 inhibitor, ECA, on the principle of symmetric and bifunctional inhibition for occupying both GST active sites [226,227]. Dabrowski and co-workers indicated that the cytotoxic effect of hypericin-based PDT was lowered in human kidney 293 cells transfected with gene-encoding GSTP1–1. Co-treatment with SX-324 and hypericin reversed the protective effects of GSTP1-1 via the increased cytotoxicity of PDT. Nevertheless, SX-324 did not modify the cytotoxicity of hypericin-based PDT in 293 cells that did not overexpress *GSTP1-1* [117].

##### Coniferyl Ferulate (Con, GST Inhibitor)

Coniferyl ferulate is a natural substance isolated from the root of *Angelica sinensis* (Oliv.) Diels. Con, showing an over 15 times stronger inhibition (IC_50_ value of 0.3 μM) of GST activity than ECA (IC_50_ value of 4.89 μM) in a high-throughput screening model using GST from the human placenta (Figure 8) [228]. Li et al. developed a drug self-delivery system made of Ce6 and Con (CeCon) as the PS and PDT enhancer, respectively. CeCon self-assembled into a nanomedicine and generated singlet oxygen similarly to Ce6-based PDT. Compared with Ce6-based PDT or Con alone, CeCon significantly decreased GST expression and activity in the A549 cells, enhancing ROS production, PDT cytotoxicity, and cell apoptosis [229].

### 3.4. HO-1 Inhibitor

#### Zn(II) Protoporphyrin IX (ZnPPIX, HO-1 Inhibitor)

Zn(II) protoporphyrin IX is a metalloporphyrin complex well-known as a selective inhibitor of HO-1, a protective enzyme against various PDT-induced cell death [230,231,232,233]. Frank and co-workers showed that the diminishing of the HO-1 activity via *HO-1* gene silencing or using ZnPPIX remarkably enhanced the effectiveness of 5-ALA-based PDT [159]. Co-treatment of ALA with ZnPPIX or *HO-1* siRNA significantly increased 5- and 2-time melanoma cell death, respectively, compared with ALA-based PDT treatment alone. Particularly, the combination of ZnPPIX and *HO-1* siRNA with ALA additionally enhanced cell death by over 6.4 times [159]. The results were confirmed in metastatic human melanoma WM451Lu cells by Grimm and co-workers. Co-treatment with ZnPPIX and ALA-based PDT significantly enhanced the cytotoxicity. Interestingly, supplementation with vitamin C at a dose of 280 μM reduced the cytotoxicity of ALA-based PDT in this model [145]. The enhancement effect of ZnPPIX was observed in Photofrin- and Talaporfin sodium-based PDT. ZnPPIX significantly increased the cytotoxicity of Photofrin in human colon adenocarcinoma C-26 and ovarian carcinoma MDAH2774 cells [142]. The cell viability of rat meningioma KMY-J cells was significantly decreased in a dose-dependent manner, followed by an increase in morphological cell damage after co-treatment with ZnPPIX and Talaporfin sodium compared with the photosensitizer alone [143]. Zhong and co-workers investigated a versatile nanoparticle-based drug delivery system for protoporphyrin IX-based PDT, containing Zn^2+^ and BSO (PZB NP) [160]. Its application significantly increased ROS generation with a decrease in the GSH level and protein level of GCS and HO-1, compared with a control group in breast cancer 4T1 cells. PZB NPs showed dose-dependent cytotoxicity, reducing 4T1 cell viability from 87% to 5% after light exposure, whereas PZ NPs (without BSO) showed slightly lower cytotoxicity (90% to 18%). The results were consistent with the in vivo model of murine breast cancer using 4T1 cells implanted to BALB/c mice. Treatment with PZB NPs significantly reduced tumor volume and weight compared with PZ NPs or NPs (without Zn^2+^ and BSO) after 14 days. The authors concluded that the nanodrug possessed dual antioxidation defense suppression properties, enhancing efficient ROS-based therapies [160]. Interestingly, ZnPPIX was also employed as an enhancer in chemodynamic treatments (CDTs). A cupric ion (Cu^2+^) was used as a CDT initiator in Cu–Zn Protoporphyrin nanoscale coordination polymers (NCPs) by converting endogenous H_2_O_2_ to a cytotoxic hydroxyl radical. ZnPPIX significantly inhibited HO-1 expression and activity in MDA-MB-231 cells. Moreover, it enhanced the cytotoxicity of the system by reducing the viability of murine breast cancer 4T1, human embryonic kidney HEK-293 T, A549 and MDA-MB-231 cells, and tumor growth in MDA-MB-231 tumor-bearing mice [234]. ZnPPIX not only acted as an HO-1 inhibitor to enhance the cytotoxicity of PDT and CDT, but could also be photoactivated and produced a cytotoxic hydroxyl radical, leading to PDT-induced cell death. Fang et al. studied the activity of polymeric ZnPPIX conjugated with the *N*-(2-hydroxypropyl)methacrylamide (pHPMA) copolymer in PDT application [235]. Polymer-conjugated Zinc Protoporphyrin showed cytotoxicity after being irradiated with 400 to 700 nm xenon light in murine sarcoma S180 model implanted into Sprague–Dawley rats and carcinogen-induced tumor models. No dark toxicity was observed in polymeric ZnPPIX-based PDT treatment [235].

## 4. Conclusions

PDT is a selective and minimal systemically toxic modality for tumor treatment that overcomes the problems generated by conventional chemotherapy in various aspects, particularly specificity and efficacy against tumor cells. Nevertheless, PDT can be limited by multiple factors, especially the alteration of the level of antioxidant molecules. At least four significant antioxidant enzymes, SOD, catalase, glutathione redox cycle, and HO-1, remarkably attenuated the effectiveness of PDT. In order to increase the sensitivity of PDT against cancer cells, antioxidant enzyme inhibitors can be employed to diminish their ROS detoxifying activity (Table 2). It is a challenge to study the use of antioxidant enzyme inhibitors to improve the efficiency of PDT in eliminating tumor cells. However, the effects of a combination of PDT with antioxidant enzyme inhibitors still needs to be studied to determine the appropriate conditions.

## Figures and Tables

**Figure 1 ijms-25-03164-f001:**
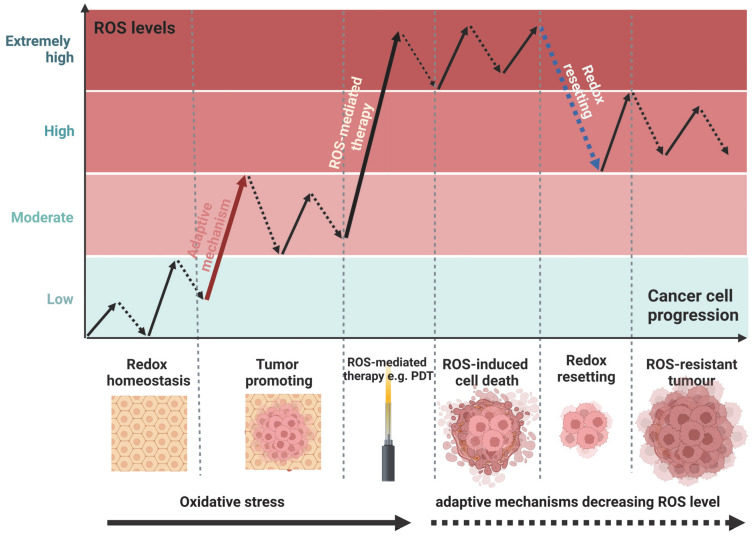
Relationship between reactive oxygen species (ROS) level and cancer cell progression. Modified from Liu et al., Gorrini et al., and Nakamura et al. [38,39,40]. Created with Biorender.com.

**Figure 2 ijms-25-03164-f002:**
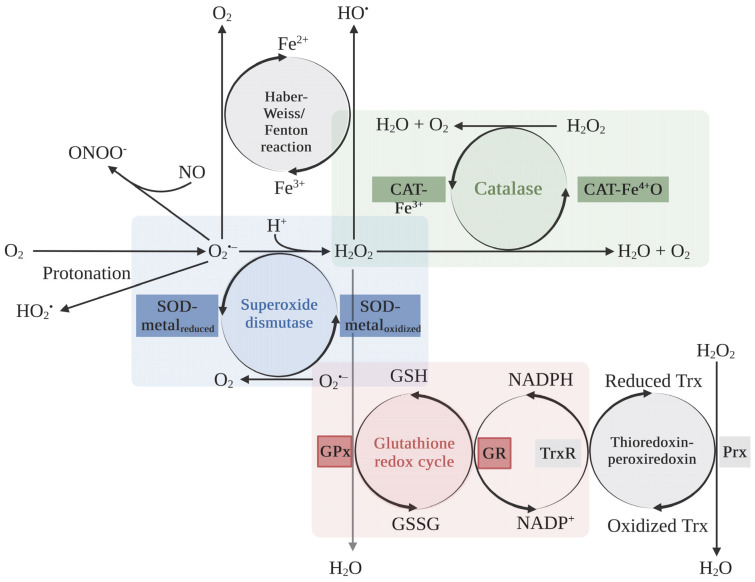
Antioxidant system: Superoxide Dismutase (SOD), Catalase (CAT), and Glutathione/Thioredoxin redox cycle enzymes. Abbreviations: CAT-Fe^3+^, ferricatalase; CAT-Fe^4+^O, compound I; GPx, glutathione peroxidase; GR, glutathione reductase; GSH, glutathione; GSSG, glutathione disulfide; H_2_O_2_, hydrogen peroxide; HO^•^, hydroxyl radical; HO_2_^•^, hydroperoxyl radical; NADPH, nicotinamide adenine dinucleotide phosphate; NO, nitric oxide; O_2_^•–^, superoxide anion; ONOO^−^, peroxynitrite; Prx, peroxiredoxins; ROS, reactive oxygen species; Trx, thioredoxin; TrxR, thioredoxin reductase [41].

**Figure 3 ijms-25-03164-f003:**
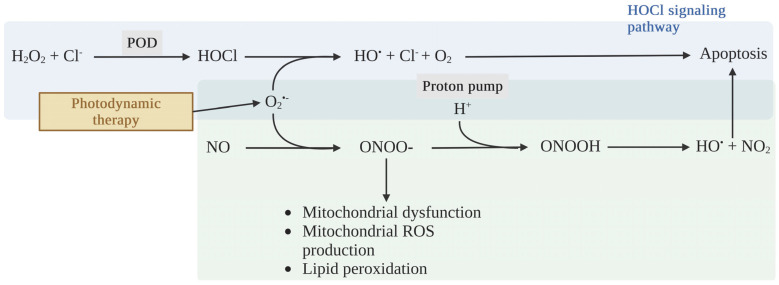
Apoptosis-inducing signaling through the hypochlorous acid (HOCl) and nitric oxide (NO)/peroxynitrite pathways [70,71].

**Figure 4 ijms-25-03164-f004:**
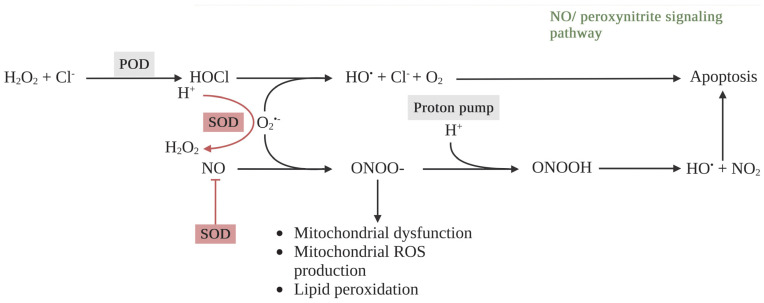
Cytoprotective mechanism of superoxide dismutase (SOD) through hypochlorous acid (HOCl) and nitric oxide (NO)/peroxynitrite signaling in tumor cells.

**Figure 5 ijms-25-03164-f005:**
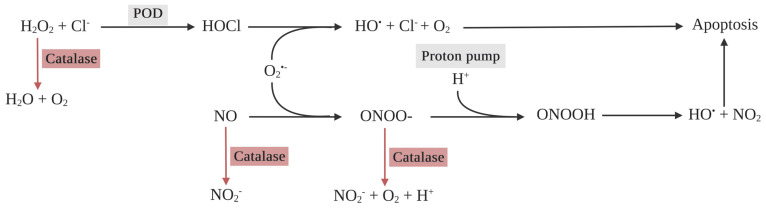
Cytoprotective mechanism of catalase through hypochlorous acid (HOCl) and nitric oxide (NO)/peroxynitrite signaling in tumor cells.

**Figure 6 ijms-25-03164-f006:**
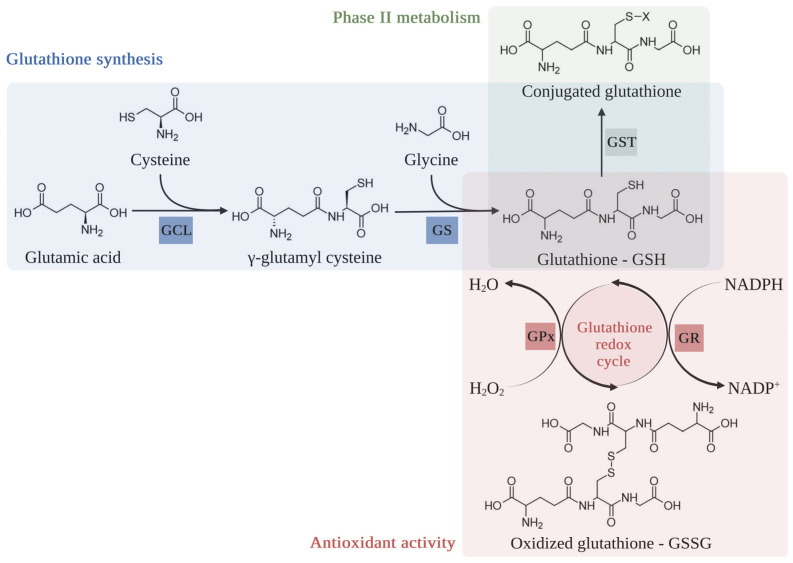
The glutathione synthesis pathway and its function as an antioxidant and detoxifying agent. Abbreviations: GCL, γ-glutamylcysteine synthetase; GS, glutathione synthetase; GR, glutathione reductase; GPx, glutathione peroxidase; GST, glutathione *S*-transferase; NADPH, nicotinamide adenine dinucleotide phosphate; X, xenobiotic.

**Figure 7 ijms-25-03164-f007:**
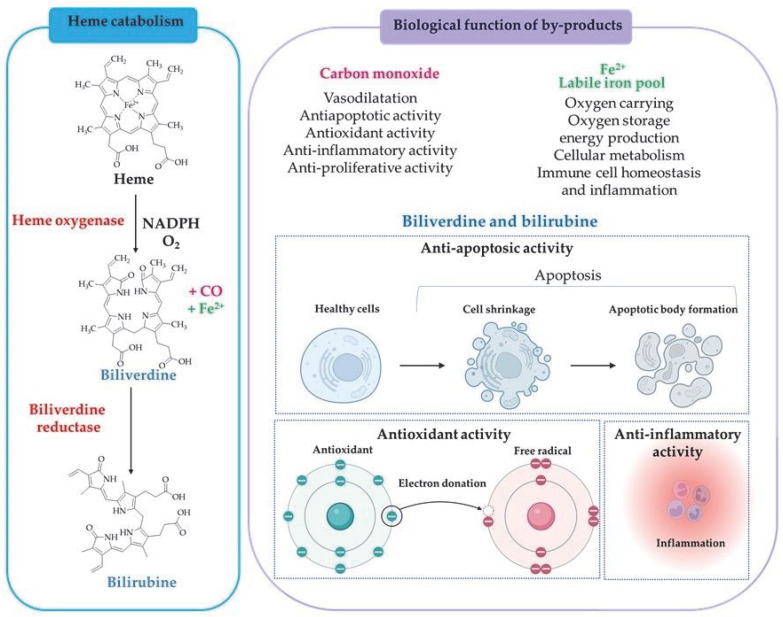
A scheme presenting the heme degradation pathway and role of heme oxygenase in producing the biologically active intermediates. Created with Biorender.com.

**Figure 8 ijms-25-03164-f008:**
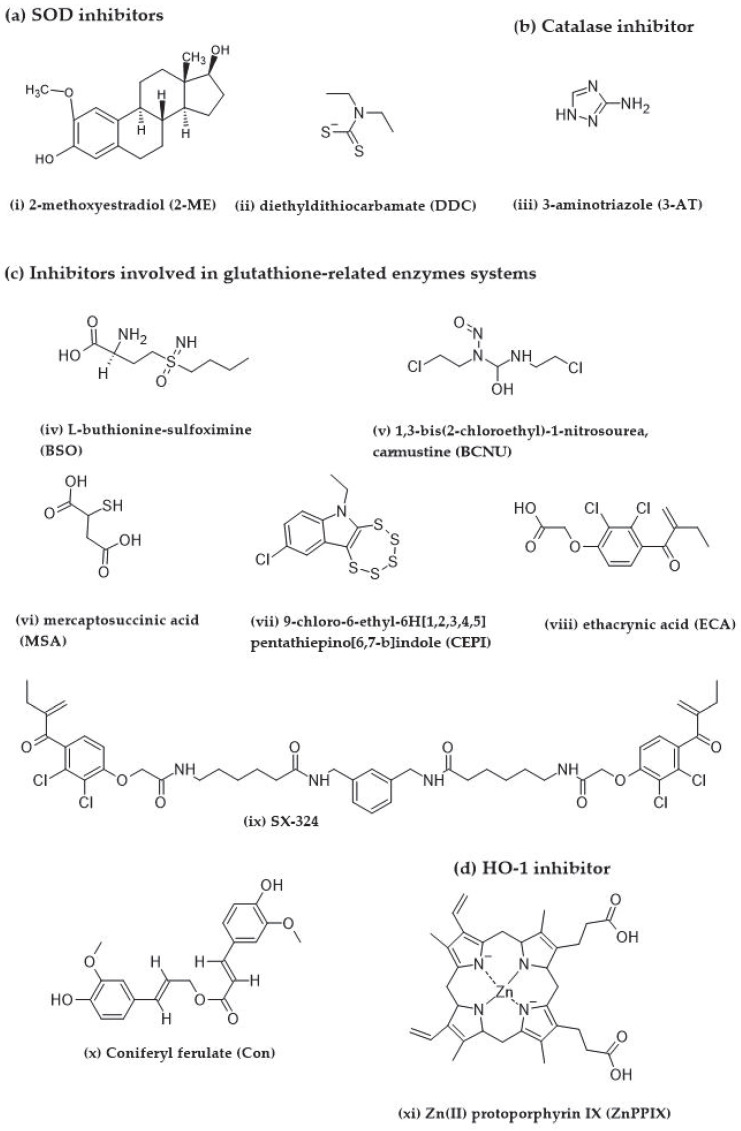
Inhibitors of antioxidant enzymes with potential applications in enhancing the effectiveness of photodynamic therapy.

**Table 2 ijms-25-03164-t002:** Antioxidant enzyme inhibitors tested in PDT.

Enzyme Inhibitor		Dose of Enzyme Inhibitor	PDT Base	Test Condition		Incubation Time (hr.)	Effectiveness of enzyme inhibitor	Ref.
Name	Target			In Vitro	In Vivo			
2-methoxy estradiol (2-ME)	SOD2	50 mM	Photofrin	Human ovarian clear carcinoma OvBh-1 cells		18	- Cell shrinkage - Actin and microtubule disruption	[176]
				Human breast adenocarcinoma MCF-7 cells				
		10 μM	Hypericin	Human breast adenocarcinoma MCF-7 cells		16	- 87.5% clonogenic ability *	[177]
		50 μM	Cyanine IR-775	Human breast adenocarcinoma MDA-MB-231 cells		24	+ 350% PDT cytotoxicity after 24 hr of irradiation* + 73.3% PDT cytotoxicity after 72 hr of irradiation *	[178]
				Human ovary adenocarcinoma SKOV-3 cells			+ 300% PDT cytotoxicity after 24 hr of irradiation * + 57.1% PDT cytotoxicity after 72 hr of irradiation *	
		3 μM	Redaporfin	Human lung adenocarcinoma A549 cells		24	+ 45.4% PDT cytotoxicity *	[78]
		0.25–10 μM	Photofrin	Murine colon adenocarcinoma C-26 cells		48	- 77.6% SOD activity after 48 hr incubation (0.5 μM) * - 87.4% SOD activity after 48 hr incubation (1 μM) * + 200% PDT cytotoxicity at 6 KJ/m^2^ (0.5 μM) *	[79]
		0.25–10 μM		Murine Lewis lung carcinoma (LLC) cells			+ 500% PDT cytotoxicity at 6 KJ/m^2^ (0.5 μM) *	
		0.06–10 μM		Murine macrophage-derived chemokine (MDC) cells			+ 1,000% PDT cytotoxicity at 5 KJ/m^2^ (0.25 μM) *	
		0.06–10 μM		Human breast cancer T47-D cells			+ 250% PDT cytotoxicity at 5 KJ/m^2^ (0.12 μM) *	
		0.25–10 μM		Human pancreatic cancer PANC-1 cells			+ 250% PDT cytotoxicity at 6 KJ/m^2^ (0.5 μM) *	
		0.06–10 μM		Human pancreatic cancer HPAF-II cells			+ 200% PDT cytotoxicity at 6 KJ/m^2^ (0.25 μM) *	
		0.25–10 μM		Human pancreatic cancer HPAC cells			+ 167% PDT cytotoxicity at 6 KJ/m^2^ (0.5 μM) *	
		0.06–10 μM		Human bladder cancer T24 cells			+ 300% PDT cytotoxicity at 6 KJ/m^2^ (0.25 μM) *	
		100 mg/Kg body weight			Murine lewis lung carcinoma (LLC) implanted into B6D2F1 mice	6 days	- 60.0% tumor volume * + Survival time	
					Murine C-26 adenocarcinoma implanted into Balb/c mice		- >90.0% tumor volume * + Survival time (60.0% cure rate)	
Diethyldithiocarbamate (DDC)	SOD1	2–16 μg/mL	Zinc phthalocyanine	Murine embryo fibroblast NIH3T3 cells		24	- 50.0% IC_50_ of the PDT	[183]
				Human breast adenocarcinoma MDA-MB-231 cells			- 50.0% IC_50_ of the PDT	
		50 μM	Meta-tetrahydroxyphenyl chlorin	Murine dorsal root ganglia; neuron cells		1.5	+ 318% PDT cytotoxicity *	[185]
		0.4 mM/Kg body weight	Photofrin II		C3H mice	2	+ 30% potentiation of ear swelling response *	[184]
		0.1 mM	Hematoporphyrin		Murine Ehrlich ascites carcinoma (EAC) implanted into mongrel mice	20mins	- 56.6% SOD1 activity - 25.5% LD_50_ of the PDT	[109]
3-aminotriazole (3-AT)	Catalase	30 mM	Benzoporphyrin	Murine leukaemia P388 cells		0.5	- 83.3% catalase activity - 23.0% LD_50_ of the PDT	[193]
		10 mM	Redaporfin	Human lung adenocarcinoma A549 cells		24	+ 20.0% PDT cytotoxicity *	[78]
		10 mM	Rose Bengal	Primary human skin fibroblasts (FB) grown in collagen gels		2	+ 16.7% PDT cytotoxicity at 150 J/cm^2^ *	[192]
		25 mM	Hematoporphyrin		Murine Ehrlich ascites carcinoma (EAC) implanted into mongrel mice	1	- 38.1% catalase activity - 21.8% LD_50_ of the PDT	[109]
		0.7 mM/Kgbody weight	Photofrin II		C3H mice	2	+ 50% potentiation of ear swelling response *	[184]
L-buthionine-sulfoximine (BSO)	GCS	0.002–10 mM	Photofrin II	Chinese hamster ovary CHO cells		1–24	- 29.0% to undetected GSH level + PDT cytotoxicity	[196]
				Chinese hamster lung V-79 cells			- 13.0% to undetected GSH level + PDT cytotoxicity	
				Murine breast carcinoma EMT-6 cells			- 19.0% to undetected GSH level + PDT cytotoxicity	
				Murine fibrosarcoma RIF-1 cells			- 12.0% to undetected GSH level + PDT cytotoxicity	
		1 mM	Hematoporphyrin	Murine leukemia L1210 cells		12	+ 3-fold log kill at 0.75 μg/mL hematoporphyrin	[197]
		1 mM	Chlorin e6 conjugated with polystyrene microsphere	Human bladder carcinoma MGH-U1 cells		18	+ 36.1% PDT cytotoxicity at 10 J/cm^2^	[198]
		600 μM	Redaporfin	Human lung adenocarcinoma A549 cells		24	+ 18.0% PDT cytotoxicity *	[78]
		100 μM	Hypericin	Human breast adenocarcinoma MCF-7 cells		Overnight	- 80.0% total GSH level	[199]
				Human breast adenocarcinoma MDA-MB-231 cells			- 80.0% total GSH level + 38.5% PDT cytotoxicity *	
		300 μM	Disulphonated aluminum phthalocyanine (AlPcS_2_)	Human breast adenocarcinoma MCF-7 cells		24	+ 34.3% PDT cytotoxicity	[175]
		500 μM	Meta-tetrahydroxyphenyl chlorin	Murine dorsal root ganglia; neuron cells		24	+ 535% PDT cytotoxicity *	[185]
				Murine satellite glia cells			+ 30.0% PDT cytotoxicity	
		0.001–10 mM	Chlorin e6	Human colorectal carcinoma HCT116 cells		24	- 78.0% GSH level at 10 μM BSO * + 45.0% PDT cytotoxicity at 0.5 μg/mL Ce6 with 10 mM BSO *	[200]
				Human ampulla vater carcinoma SNU478 cells			- 66.7% GSH level (10 μM BSO) + 72.7% PDT cytotoxicity at 0.5 μg/mL Ce6 with 10 mM BSO *	
		3 mM	Aluminum (III) phthalocyanine tetrasulfonate (AlPcS_4_)	Human epidermoid carcinoma A431 cells		18	- 83.3% GSH level + 144% PDT cytotoxicity at 2 J/cm^2^ *	[195]
		0.5–1000 μg/mL	Photofrin	Human glioma U87 cells: and U251n cells		24	+ 70.0% PDT cytotoxicity at 5 μg/mL Photofrin with 0.5 μg/mL *	[201]
				Human glioma U251n cells			+ 60.0% PDT cytotoxicity at 5 μg/mL Photofrin with 0.5 μg/mL *	
		440 mg/Kg body weight			Human U87 glioblastoma implanted into rats		+ Superficial tumor damage + 114.3% lesion volume at 70 J/cm^2^	
		440 mg/Kg body weight	Photofrin		Murine 9L gliosarcoma implanted into Fischer rats	24	- 67.1% GSH level + 111.1% lesion volume	[202]
		10 mM	Chlorin e6-loaded poly(ethylene glycol)-block-poly(D,L lactide) nanoparticles	Murine carcinoma SCC-7 cells		28	- 75.6% GSH level + 50.0% PDT cytotoxicity at 2 μg/mL Ce6 *	[203]
		3 mmol/Kg body weight			Murine carcinoma SCC-7 into implanted mice	12	- 75% tumor size after 14 days * + Apoptosis and tissue damage	
		N/A	Indocyanine green in near-infrared (NIR) photothermal liposomal nanoantagonists	Murine breast cancer 4T1 cells		24	- 26.4% GSH level + 90.9% PDT cytotoxicity + 1.5-fold ROS level	[204]
					Murine breast cancer 4T1 implanted into BALB/c mice	18–24	- 2.9-fold GSH level - 2.0-fold tumor weight	
		4 mM/Kg body weight	Hematoporphyrin		Murine Ehrlich ascites carcinoma (EAC) implanted into mongrel mice	14	- 68.2% GSH level - 63.6% total glutathione level - 21.5% LD_50_ of the PDT	[109]
		0.2–4 μM	Protoporphyrin IX	Murine breast cancer 4T1 cells		8	- 28.0% GSH level at 4 μM BSO - GCS expression at 4 μM BSO + 50.0% PDT cytotoxicity at 0.8 μM BSO	[160]
		100 μL of 2 mM			Murine breast cancer 4T1 implanted into BALB/c mice	4	- 50.0% tumor volume after 12 days * - 62.5% tumor weight after 12 days *	
1,3-bis(2-chloroethyl)-1-nitrosourea (BCNU)	GR	500 μM	Aluminum (III) phthalocyanine tetrasulfonate (AlPcS4)	Human epidermoid carcinoma A431 cells		1	+ 23.4% ROS level at 1.8 J/cm^2^ continuous light * + 146% PDT cytotoxicity at 2.25 J/cm^2^ continuous light *	[211]
			Hypericin				+ 23.5% ROS level at 0.162 J/cm^2^ continuous light * + 38.5% PDT cytotoxicity at 0.2 J/cm^2^ continuous light *	
		500 μM	Aluminum (III) phthalocyanine tetrasulfonate (AlPcS_4_)	Human epidermoid carcinoma A431 cells		1	+ 35.3% PDT cytotoxicity at 2 J/cm^2 *^	[195]
		50 μM	Photofrin	Ad*MnSOD* transfected human breast carcinoma ZR-75-1 cells		1	+ 120% PDT cytotoxicity	[212]
		100 μM	Hypericin	Human breast adenocarcinoma MCF-7 cells		Overnight	+ 100% PDT cytotoxicity in combination with 100 μM BSO *	[199]
		0.1 mM	Hematoporphyrin		Murine Ehrlich ascites carcinoma (EAC) implanted into mongrel mice	25 mins	- 64.2% GR activity - 21.8% LD_50_ of the PDT	[109]
Mercaptosuccinic acid (MSA)	GPx1	1.5 mM	Rose Bengal	Primary human skin fibroblasts (FB) grown in collagen gels		2	+ 33.3% PDT cytotoxicity at 150 J/cm^2^ *	[192]
		10 mM	Chlorin e6	Human extrahepatic cholangiocarcinoma SNU1196 cells		0.5	- 58.1% GPx activity * + 60.0% ROS level * + 60.0% PDT cytotoxicity *	[122]
		1–700 μmol/L	meta-tetrahydroxyphenyl chlorin	Human lung carcinoma A-427 cells		24	+ Synergistic effect (CI < 1)	[217]
				Human oral carcinoma BHY cells				
				Human esophageal carcinoma KYSE-70 cells				
				Human urinary bladder carcinoma RT-4 cells				
9-chloro-6-ethyl-6*H*[1,2,3,4,5]pentathiepino[6,7-b]indole (CEPI)	GPx1	0.01–50 μmol/L	meta-tetrahydroxyphenyl chlorin	Human esophageal carcinoma KYSE-70 cells		24	+ Synergistic effect (CI < 1)	[217]
		4.0–15.9 μmol/L		Human urinary bladder carcinoma RT-4 cells				
Ethacrynic acid (ECA)	GSTP1-1	5 μM	Ethacrynic acid-conjugated brominated BODIPY	Human breast adenocarcinoma MCF-7 cells		12	+ 50.0% PDT cytotoxicity *	[225]
				Human breast adenocarcinoma MDA-MB-231 cells			+ 133% PDT cytotoxicity* + 36.4% ROS level* + 7.14% Singlet oxygen level * + 13.1% Superoxide anion level * + 243% Hydroxyl radical and peroxynitrite anion level* - 15% GSH level without irradiation* - 730% GSH level with irradiation *	
		5 mg/Kg body weight			Human breast adenocarcinoma MDA-MB-231 implanted into immunodeficient nude mice	6	- 56.9% tumor volume *	
SX-324	GSTP1-1	1 μM	Hypericin	GSTP1-1- overexpressed human kidney fibroblast K293 cells		1	+ 87.2% PDT cytotoxicity	[117]
Coniferyl ferulate (Con)	GST	0.2–2.7 mg/L	Drug self-delivery systems of chlorin e6 and coniferyl ferulate	Human lung adenocarcinoma A549 cells		20	+ 250% ROS level* + 200% PDT cytotoxicity at 2.7 mg/mL *	[229]
Zn(II) protoporphyrin IX (ZnPPIX)	HO-1	400 μM	5-aminolevulinic acid	Human melanoma WM451Lu cells		16	+ 499% PDT cytotoxicity + 641% PDT cytotoxicity (combined with HO-1 siRNA)	[159]
		5 μM	5-aminolevulinic acid	Human melanoma WM451Lu cells		16	+ 100% PDT cytotoxicity *	[145]
		1.25–2.5 μM	Photofrin	Murine colon adenocarcinoma C-26 cells		24	+ 77.5% PDT cytotoxicity at 4.5 KJ/m^2^	[142]
				Human ovarian carcinoma MDAH2774 cells			+ >42.8% PDT cytotoxicity at 4.5 KJ/m^2^	
		1 μM	Talaporfin sodium	Murine meningioma KMY-J cells		4	+ 900% PDT cytotoxicity at 19.2 μM talaporfin sodium * + Morphological cell damage	[143]
		2 μM	Protoporphyrin IX	Murine breast cancer 4T1 cells		8	- HO-1 expression	[160]
		2.81 mg/Kg body weight			Murine breast cancer 4T1 implanted intoBALB/c mice	4	- 70.0% tumor volume after 12 days * - 41.8% tumor weight after 12 days *	

Note. GCS, γ-glutamylcysteine synthetase; GPx, glutathione peroxidase; GR, glutathione reductase; GST, glutathione *S*-transferase; SOD, superoxide dismutase; *, estimated from the figure of references; N/A, no available data.

## Data Availability

Not applicable.

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
