# Peer review of "Antioxidant Enzymes in Cancer Cells: Their Role in Photodynamic Therapy Resistance and Potential as Targets for Improved Treatment Outcomes"

_ijms, 2024, doi:10.3390/ijms25063164_

Round 1
Reviewer 1 Report
Comments and Suggestions for Authors
The work entitled “Antioxidant Enzymes in Cancer Cells: Their Role in Photodynamic Therapy Resistance and Potential as Targets for Improved Treatment Outcomes” by Udomsak et al extensively reviews the existing literature on the modulation of photodynamic effects employing the ability of antioxidant enzymes and their inhibitors. The Ms appears to be interesting for Int. J. Mol. Sci. readers.
The review is quite comprehensive, and its reading can be challenging. I recommend grammatical corrections by a native English speaker. The final table summarizing the different uses of the enzyme inhibitors is greatly acknowledged.
Minor points:
Lines 84 to 86 “Nonetheless, the optical window to penetrate the tissue in PDT has been limited to only between 600 and 800 nm (red to deep red). More than 800 nm of light does not provide enough energy to activate PS and produce ROS. At the same time, up to 600 nm light could be absorbed by water,”
I´d say that water absorbs at wavelengths higher than 800 nm, and up to 600 nm, hemoglobin, melanin and other molecules may be absorbing, Please check the literature.
Figure 7: apoptotic instead of apoptosic
The action of sulfide species on the outcome of PDT such as demonstrated in Calvo et al. (Nitric Oxide. 2022 Aug 1;125-126:57-68. doi: 10.1016/j.niox.2022.06.006.), should be interesting to discuss.
Comments on the Quality of English Language.I recommend grammatical corrections by a native English speaker.
Author Response
Antioxidant Enzymes in Cancer Cells: Their Role in Photodynamic Therapy Resistance and Potential as Targets for Improved Treatment Outcomes (ijms-2896535)
International Journal of Molecular Sciences (ISSN 1422-0067)
March 3rd, 2024
Dear Reviewer,
I am writing to express my sincere gratitude for your time and effort in reviewing our manuscript titled "Antioxidant Enzymes in Cancer Cells: Their Role in Photodynamic Therapy Resistance and Potential as Targets for Improved Treatment Outcomes". Your insightful comments and constructive feedback have been immensely valuable in refining our work.
We have carefully considered each of the points raised in your review and have made corresponding revisions to address your highlighted concerns. Below, I outline how we have incorporated your feedback into the revised version of our manuscript:
- Regarding the absorption wavelength of chromophores in tissues, there are cytochromes, melanin, water, deoxyhemoglobin, and oxyhemoglobin show absorption wavelengths close to 600 to 800 nm. To clarify this, we included sentences in red letters in lines 91 to 96.
“The absorption spectra of significant chromophores in tissues, i.e., cytochromes, melanin, water, deoxyhemoglobin, and oxyhemoglobin, were reported by Plaetzer’s group [26]. It is important to note that hemoglobin and oxyhemoglobin exhibit similar absorption wavelengths as utilized by some photosynthesizes used in PDT. Certain studies suggest that PDT may elevate the production of methemoglobin and deoxygenated hemoglobin, particularly under conditions of hemolysis. [27-30]”
This paragraph is supported by the following citations:
- Plaetzer, K.; Krammer, B.; Berlanda, J.; Berr, F.; Kiesslich, T. Photophysics and Photochemistry of Photodynamic Therapy: Fundamental Aspects. Lasers Med Sci 2009, 24, 259–268, doi:10.1007/s10103-008-0539-1.
- Heckl, C.; Aumiller, M.; Rühm, A.; Sroka, R.; Stepp, H. Fluorescence and Treatment Light Monitoring for Interstitial Photodynamic Therapy. Photochem & Photobiology 2020, 96, 388–396, doi:10.1111/php.13203.
- Teles De Andrade, C.; Nogueira, M.S.; Kanick, S.C.; Marra, K.; Gunn, J.; Andreozzi, J.; Samkoe, K.S.; Kurachi, C.; Pogue, B.W. Optical Spectroscopy of Radiotherapy and Photodynamic Therapy Responses in Normal Rat Skin Shows Vascular Breakdown Products.; Kessel, D.H., Hasan, T., Eds.; San Francisco, California, United States, March 1 2016; p. 969410.
- Larsen, E.L.P.; Randeberg, L.L.; Gederaas, O.A.; Arum, C.-J.; Hjelde, A.; Zhao, C.-M.; Chen, D.; Krokan, H.E.; Svaasand, L.O. Monitoring of Hexyl 5-Aminolevulinate-Induced Photodynamic Therapy in Rat Bladder Cancer by Optical Spectroscopy. J. Biomed. Opt. 2008, 13, 044031, doi:10.1117/1.2967909.
- Hamada, R.; Ogawa, E.; Arai, T. Continuous Optical Monitoring of Red Blood Cells During a Photosensitization Reaction. Photobiomodulation, Photomedicine, and Laser Surgery 2019, 37, 110–116, doi:10.1089/photob.2018.4513.
- There was a misspelled word in Figure 7 in line 447. I have changed the word “Anti-apoptosic activity” to “Anti-apoptotic activity”
- Regarding the action of sulfide species on the outcome of PDT, we found that hydrogen sulfide can reduce the activity of PDT by increasing GSH levels and catalase activity. We included sentences in red letters in lines 431 to 435.
“Interestingly, hydrogen sulfide (H2S) has been observed to diminish the activity of PDT-based therapy by ROS/ RNS scavenging. The cytotoxicity of 5-ALA was dramatically reduced following exposure to H2S in murine adenocarcinoma LM2 cell line. The outcomes were associated with an elevation of GSH levels and catalase activity with reduction of singlet oxygen level [127].”
This paragraph is supported by the following citation:
- Calvo, G.; Céspedes, M.; Casas, A.; Di Venosa, G.; Sáenz, D. Hydrogen Sulfide Decreases Photodynamic Therapy Outcome through the Modulation of the Cellular Redox State. Nitric Oxide 2022, 125–126, 57–68, doi:10.1016/j.niox.2022.06.006.
According to the Reviewer's suggestion, we have also corrected grammatical errors in our manuscript.
Once again, thank you for your kindness and contribution to improving our manuscript.
Kind regards,
Prof. Marek Murias, PhD, DSc
Reviewer 2 Report
Comments and Suggestions for Authors
Reviewer comments
The present review article discusses the role of photodynamic therapy and the improvement of the current strategy used in cancer treatment and management. PDT is the most common and effective strategy in cancer treatment; however, the generation of reactive species generated by PDT is diminished by antioxidant enzymes found in cancer cells, which reduces the efficiency and efficacy of PDT. The use of antioxidant inhibitors with PDT could help to improve the effectiveness of PDT by diminishing the antioxidant activities of tumor cells.
Recommendation
1. The paper is scientifically sound, well-planned, and written in an organized manner.
2. The aim of the review article is well discussed and supported with previously published data.
3. Figures and tables are nicely compiled and support the review article.
4. The references are appropriate and sufficient.
Scientific comments
1. In Table 1, add one more column for the name of the inhibitor target. Few inhibitors are discussed in the text; however, few are not.
2. Add 1 table of antioxidant enzymes showing their location, target, function, and how they affect the current strategy or therapy used in cancer treatment.
3. Figure 1, Enhance the resolution of the figure. The word-adaptive mechanism is not clear.
4. Figure 7, uses a high-resolution figure.
Minor comments/typo errors
1. Delete full stop from the title.
2. Delete the word 'and' after citation 134 from line 479.
Author Response
Antioxidant Enzymes in Cancer Cells: Their Role in Photodynamic Therapy Resistance and Potential as Targets for Improved Treatment Outcomes (ijms-2896535)
International Journal of Molecular Sciences (ISSN 1422-0067)
March 3rd, 2024
Dear Reviewer,
I am writing to express my sincere gratitude for your time and effort in reviewing our manuscript titled "Antioxidant Enzymes in Cancer Cells: Their Role in Photodynamic Therapy Resistance and Potential as Targets for Improved Treatment Outcomes". Your insightful comments and constructive feedback have been immensely valuable in refining our work.
We have carefully considered each point raised in your review and made corresponding revisions to address your highlighted concerns. Below, we outline in italics how we have incorporated your feedback into the revised version of our manuscript:
Scientific comments
- In Table 1, add one more column for the name of the inhibitor target. Few inhibitors are discussed in the text; however, few are not.
Regarding the comments on Table 2 (previously Table 1) in line 865, we agree that we should add one column describing the target of the antioxidant enzyme inhibitors. It’s next to name the antioxidant enzyme inhibitors.
- Add 1 table of antioxidant enzymes showing their location, target, function, and how they affect the current strategy or therapy used in cancer treatment.
Thank you for this suggestion, we have prepared an additional table, Table 1, which starts in line 513. It includes the target of action, location, function, and therapeutic effects on cancers of each antioxidant enzyme inhibitor.
Minor comments/typo errors
Figure 1, Enhance the resolution of the figure. The word-adaptive mechanism is not clear.
Figure 7, uses a high-resolution figure.
We have prepared and included a new version of Figure 1 in line 113 and Figure 7 in line 447.
- Delete full stop from the title.
- Delete the word 'and' after citation 134 from line 479.
Thank you very much, we have deleted the full stop from the title and the word “and” in line 490.
Please find attached the revised manuscript. I welcome any further suggestions or feedback you may have and am committed to addressing any remaining concerns to ensure the highest standard of excellence.
Once again, thank you for your invaluable contribution to developing our manuscript.
With kind regards,
Prof. Marek Murias, PhD, DSc
Reviewer 3 Report
Comments and Suggestions for Authors
The presented review is devoted to antioxidant enzymes in cancer cells and their role in photodynamic therapy resistance because of strong reprogramming of these enzymes in the tumor environment. Besides, the various strategies of tumor cell antioxidant activity suppression have been also considered. Authors provide the extensive information on various antioxidant inhibitors which enable, in principle, to improve the effectiveness of PDT . Generally, this article should be quite useful for the specialists dealing with the problem of anticancer therapy resistance. However, in my opinion, it would be good for authors to mention some relatively new and important mechanism which indicated that majority of cancer cell lines were resistant to GSH depletion. [ Harris IS et al , Deubiquitinases maintain protein homeostasisand survival of cancer cells upon glutathione depletion. // Cell Metabolism (2019) 29:1166–81 doi: 10.1016/j.cmet.2019.01.020 ]. Inhibition of GSH synthesis is effective only in combination with inhibition of deubiquitinating ensymes (DUBs). This realy leads to the strong accumulation of polyubiquitinated proteins, induction of proteotoxic stress and cancer cell death. So depletion of GSH renders cancer cells strongly dependent on DUBs activity to maintain protein homeostasis and cancer cell viability and reveal a potentially exploitable vulnerability for cancer therapy. In line with this, an inefficiency of GSH depletion with BSO may be due to the overexpression of deubiquitinases that inhibit protein degradation following ER-stress.
Author Response
Antioxidant Enzymes in Cancer Cells: Their Role in Photodynamic Therapy Resistance and Potential as Targets for Improved Treatment Outcomes (ijms-2896535)
International Journal of Molecular Sciences (ISSN 1422-0067)
March 3rd, 2024
Dear Reviewer,
I am writing to express my sincere gratitude for your time and effort in reviewing our manuscript titled "Antioxidant Enzymes in Cancer Cells: Their Role in Photodynamic Therapy Resistance and Potential as Targets for Improved Treatment Outcomes". Your insightful comments and constructive feedback have been immensely valuable in refining our work.
We have carefully considered each point raised in your review and made corresponding revisions to address your highlighted concerns. Below, I outline how we have incorporated your feedback into the revised version of the literature review:
- Regarding the mechanism of cancer cell resistance to GSH depletion, the effects of deubiquitinase inhibition are described in line 704 in red letters.
“In addition of GSH depletion by BSO treatment, inhibition of deubiquitinating enzymes substantially interfere with cancer cell survival upon oxidative stress. Harris et al. indicated that inhibition of deubiquitinating enzymes in combination with BSO treatment increased cell death by ferroptosis, a cell death pathway activated by excessive lipid peroxidation. [206] Therefore, the cytotoxicity of PDT-based therapy is feasibly enhanced by deubiquitinating enzyme inhibitors combined with BSO.”
This paragraph was supported by the following citation:
- Harris, I.S.; Endress, J.E.; Coloff, J.L.; Selfors, L.M.; McBrayer, S.K.; Rosenbluth, J.M.; Takahashi, N.; Dhakal, S.; Koduri, V.; Oser, M.G.; et al. Deubiquitinases Maintain Protein Homeostasis and Survival of Cancer Cells upon Glutathione Depletion. Cell Metabolism 2019, 29, 1166-1181.e6, doi:10.1016/j.cmet.2019.01.020.
Please find attached the revised manuscript. I welcome any further suggestions or feedback you may have and am committed to addressing any remaining concerns to ensure the highest standard of excellence.
Once again, thank you for your invaluable contribution to developing our manuscript.
Sincerely yours,
Prof. Marek Murias, PhD, DSc
Reviewer 4 Report
Comments and Suggestions for Authors
General comment:
This review manuscript, entitled “Antioxidant Enzymes in Cancer Cells: Their Role in Photodynamic Therapy Resistance and Potential as Targets for Improved Treatment Outcomes,” authored by Udomsak et al., surveyed the cytoprotective mechanism of detoxifying ROS enzymes that interfere with PDT-induced cell death and antioxidant inhibitors as a strategy to diminish the antioxidant activities of tumor cells and improve the effectiveness of PDT. The author put fantastic effort into accumulating valuable and suitable information for publication in the International Journal of Molecular Sciences. I have a few questions which should be addressed before publication.
Specific comments:
1) How good is PDT in the case of an anemic person?
2) Whether PDT treatment affects all heme proteins and enzymes as of redox properties of these heme proteins?
3) How many detectable doses of light are good for noninvasive damage to the skin?
4) How do cytochromes P450 enzymes get affected because of this treatment that may lead to another alteration of the metabolic pathway?
Comments on the Quality of English LanguageEnglish is fine.
Author Response
Antioxidant Enzymes in Cancer Cells: Their Role in Photodynamic Therapy Resistance and Potential as Targets for Improved Treatment Outcomes (ijms-2896535)
International Journal of Molecular Sciences (ISSN 1422-0067)
March 3rd, 2024
Dear Reviewer,
I am writing to express my sincere gratitude for your time and effort in reviewing our manuscript titled "Antioxidant Enzymes in Cancer Cells: Their Role in Photodynamic Therapy Resistance and Potential as Targets for Improved Treatment Outcomes". Your insightful comments and constructive feedback have been immensely valuable in refining my work.
I have carefully considered each point raised in your review and made corresponding revisions to address your highlighted concerns. Below, we outline in italics our replies:
- How good is PDT in the case of an anemic person?
Due to limited evidence, it’s difficult to conclude how good PDT works on anemic person. There are two articles indicating that PDT can still show its activity in anemia-related patients.
A group of Pavlič reported a case of a 16-year-old girl who was diagnosed with a periodontal disease associated with Fanconi anemia. The patient was treated by PDT using phenothiazine chloride as a photosensitizer every 2 months for 10 months. The results indicated that PDT effectively reduced periodontopathogenic bacteria, leading to clinical improvement in plaque index, gingival index, probing pocket depth, and bleeding on probing (Pavlič et al., 2017).
Al Asmari et al. showed the adjunctive role of PDT with full-mouth ultrasonic scaling (FMUS) in 30 patients diagnosed with thalassemia major (TM) patients with gingivitis. Group A patients treated with PDT, methylene blue (Helbo Blue photosensitizer) with FMUS showed significant improvement in plaque and gingival index after 12 weeks compared to group B patients treated with FMUS alone (Al Asmari and Khan, 2020).
However, Golab’s group reported the opposite results in vivo experiment. In the model of murine colon-26 adenocarcinoma model syngeneic with BALB/c mice, acute hemolytic anemia, and chemotherapy-induced anemia were induced by the administration of phenylhydrazine hydrochloride and carboplatin, respectively. The effectiveness of photofrin-based therapy was significantly diminished in both phenylhydrazine hydrochloride- and carboplatin-induced anemia mice. Importantly, the administration of recombinant human erythropoietin can reverse sensitivity of PDT in carboplatin-induced anemia mice (Gołab et al., 2002).
- Whether PDT treatment affects all heme proteins and enzymes as of redox properties of these heme proteins?
It is possible that PDT can affects to heme proteins and enzymes which show absorption wavelengths close to 600 to 800 nm. To clarify this, I included sentences in red letters in lines 91 to 96.
“The absorption spectra of significant chromophores in tissues, i.e., cytochromes, melanin, water, deoxyhemoglobin, and oxyhemoglobin, were reported by Plaetzer’s group [26]. It is important to note that hemoglobin and oxyhemoglobin exhibit identical absorption wavelengths as utilized in PDT. Certain studies suggest that PDT may elevate the production of methemoglobin and deoxygenated hemoglobin, particularly under conditions of hemolysis [27-30].”
This paragraph is supported in the manuscript by the following citations:
- Plaetzer, K.; Krammer, B.; Berlanda, J.; Berr, F.; Kiesslich, T. Photophysics and Photochemistry of Photodynamic Therapy: Fundamental Aspects. Lasers Med Sci 2009, 24, 259–268, doi:10.1007/s10103-008-0539-1.
- Heckl, C.; Aumiller, M.; Rühm, A.; Sroka, R.; Stepp, H. Fluorescence and Treatment Light Monitoring for Interstitial Photodynamic Therapy. Photochem & Photobiology 2020, 96, 388–396, doi:10.1111/php.13203.
- Teles De Andrade, C.; Nogueira, M.S.; Kanick, S.C.; Marra, K.; Gunn, J.; Andreozzi, J.; Samkoe, K.S.; Kurachi, C.; Pogue, B.W. Optical Spectroscopy of Radiotherapy and Photodynamic Therapy Responses in Normal Rat Skin Shows Vascular Breakdown Products.; Kessel, D.H., Hasan, T., Eds.; San Francisco, California, United States, March 1 2016; p. 969410.
- Larsen, E.L.P.; Randeberg, L.L.; Gederaas, O.A.; Arum, C.-J.; Hjelde, A.; Zhao, C.-M.; Chen, D.; Krokan, H.E.; Svaasand, L.O. Monitoring of Hexyl 5-Aminolevulinate-Induced Photodynamic Therapy in Rat Bladder Cancer by Optical Spectroscopy. J. Biomed. Opt. 2008, 13, 044031, doi:10.1117/1.2967909.
- Hamada, R.; Ogawa, E.; Arai, T. Continuous Optical Monitoring of Red Blood Cells During a Photosensitization Reaction. Photobiomodulation, Photomedicine, and Laser Surgery 2019, 37, 110–116, doi:10.1089/photob.2018.4513.
- How many detectable doses of light are good for noninvasive damage to the skin?
The optimal dose of light varies widely depending on the specific condition being treated, the type of light used (e.g., LED, laser, UV), and the individual's skin type and sensitivity. For example, treatments like UVB phototherapy for psoriasis or eczema are carefully dosed to minimize the risk of skin damage while effectively reducing symptoms. Similarly, low-level laser therapy (LLLT) or LED light treatments for acne or skin rejuvenation are designed to deliver beneficial effects without harming the skin.
It's important to note that there is no one-size-fits-all answer for the number of "detectable doses" of light that are beneficial for the skin. Treatment protocols are highly individualized and are based on a careful assessment by healthcare professionals. Overexposure to certain types of light, particularly UV light, can indeed cause skin damage, including burns, premature aging, and increased risk of skin cancer. Therefore, noninvasive light treatments should always be administered under the guidance of a qualified practitioner who can determine the appropriate dosage and frequency for each individual's needs.
According to a phase I clinical trial of 17 patients diagnosed with carcinoma-insitu (CIS) and microinvasive cancer (MIC) of the central airways by Dhillon and coworkers the authors recommended the maximum light dose at 125 J/cm2. The patients were treated by fixed doses of PDT, 2-[1hexyloxyethyl]-2-devinyl pyropheophorbide-a (HPPH) with light dose escalation starting from 75 J/cm2 to 85, 95, 125, and 150 J/cm2 respectively, then evaluated the toxicity of PDT after 1 week, 1 month and 6 months of treatment. A serious adverse event was shown in one of the three patients in 150 J/cm2 light dose group (Dhillon et al., 2016). However, the maximum light dose also depends on sensitivity of the skin. Davidson’s group reported that patients with prostate cancer experienced rectal damage with a maximum rectal dose at 10 J/cm2 (Davidson et al., 2009).
- How do cytochromes P450 enzymes get affected because of this treatment that may lead to another alteration of the metabolic pathway?
Generally, the interactions of photosensitizers with cytochrome P450 enzymes are still not fully explored and similar spotrzeżenie can be made regarding transmembrane transporters. An intriguing study in this context, related to CYP3A4, has been published by Semelakova and co-workers. Their research demonstrates that treatment with hyperforin or aristoforin, under both light and dark PDT conditions, significantly impacted CYP3A4 activity and inhibited the functionality of specific membrane transport proteins in HT-29 cells. Additionally, it was discovered that hyperforin or aristoforin had a significant effect on the intracellular accumulation of hypericin in HT-29 colon adenocarcinoma cells. These findings imply that hypericin, hyperforin, and aristoforin could alter the efficacy of anti-cancer medications through interactions with CYP3A4 and membrane transporters (Šemeláková et al., 2016).
References:
Al Asmari, D., Khan, M.K., 2020. Effect of photodynamic therapy on gingival inflammation in patients with thalassemia. Photodiagnosis Photodyn. Ther. 29, 101595. https://doi.org/10.1016/j.pdpdt.2019.101595
Davidson, S.R.H., Weersink, R.A., Haider, M.A., Gertner, M.R., Bogaards, A., Giewercer, D., Scherz, A., Sherar, M.D., Elhilali, M., Chin, J.L., Trachtenberg, J., Wilson, B.C., 2009. Treatment planning and dose analysis for interstitial photodynamic therapy of prostate cancer. Phys. Med. Biol. 54, 2293–2313. https://doi.org/10.1088/0031-9155/54/8/003
Dhillon, S.S., Demmy, T.L., Yendamuri, S., Loewen, G., Nwogu, C., Cooper, M., Henderson, B.W., 2016. A Phase I Study of Light Dose for Photodynamic Therapy Using 2-[1-Hexyloxyethyl]-2 Devinyl Pyropheophorbide-a for the Treatment of Non–Small Cell Carcinoma In Situ or Non–Small Cell Microinvasive Bronchogenic Carcinoma: A Dose Ranging Study. J. Thorac. Oncol. 11, 234–241. https://doi.org/10.1016/j.jtho.2015.10.020
Gołab, J., Olszewska, D., Mroz, P., Kozar, K., Jakobisiak, M., 2002. Erythropoietin Restores the Antitumor Effectiveness of Photodynamic Therapy in Mice with Chemotherapy- induced Anemia. Clin. Cancer Res. 8, 1265–1270.
Pavlič, A., Matoh, U., Rajić, V., Petelin, M., 2017. Effect of Repeated Antimicrobial Photodynamic Therapy in Treatment of Periodontitis Associated with Fanconi Anemia. Photomed. Laser Surg. 35, 64–68. https://doi.org/10.1089/pho.2016.4122
Šemeláková, M., Jendželovský, R., Fedoročko, P., 2016. Drug membrane transporters and CYP3A4 are affected by hypericin, hyperforin or aristoforin in colon adenocarcinoma cells. Biomed. Pharmacother. Biomedecine Pharmacother. 81, 38–47. https://doi.org/10.1016/j.biopha.2016.03.045
Please find attached the revised manuscript. We welcome any further suggestions or feedback you may have and we are committed to addressing any remaining concerns to ensure the highest standard of excellence.
Once again, thank you for your precious contribution to developing our manuscript.
Sincerely yours,
Prof. Marek Murias, PhD, DSc